# Large Pretraining Datasets Don't Guarantee Robustness after Fine-Tuning

## Abstract

Large-scale pretrained models are widely leveraged as foundations for learning new specialized tasks via fine-tuning, with the goal of maintaining the general performance of the model while allowing it to gain new skills. A valuable goal for all such models is robustness: the ability to perform well on out-of-distribution (OOD) tasks. We assess whether fine-tuning preserves the overall robustness of the pretrained model, and observed that models pretrained on large datasets exhibited strong catastrophic forgetting and loss of OOD generalization. To systematically assess robustness preservation in fine-tuned models, we propose the Robustness Inheritance Benchmark (ImageNet-RIB). The benchmark, which can be applied to any pretrained model, consists of a set of related but distinct OOD (downstream) tasks and involves fine-tuning on one of the OOD tasks in the set then testing on the rest. We find that though continual learning methods help, fine-tuning reduces robustness across pretrained models. Surprisingly, models pretrained on the largest and most diverse datasets (e.g., LAION-2B) exhibit both larger robustness losses and lower absolute robustness after fine-tuning on small datasets, relative to models pretrained on smaller datasets. We observe this collapse in contrastively pretrained (CLIP) models, where it grows with pretraining scale; the supervised models we test do not exhibit it. These findings suggest that starting with the strongest foundation model is not necessarily the best approach for performance on specialist tasks.

## 1 Introduction

Deep learning has moved toward training large models with deeper architectures (Dosovitskiy et al., 2021; He et al., 2016; Jiang et al., 2023) on massive datasets (Lin et al., 2014; Russakovsky et al., 2015; Schuhmann et al., 2022). These models exhibit impressive performance and generalization abilities; as a result, it has become common to leverage these models as a foundation for fine-tuning on specific downstream datasets to achieve better performance than training from scratch. Fine-tuning can be done with modest amounts of data, and thus is an attractive approach in applications where not enough data is available.

While this approach capitalizes on the extensive knowledge embedded in pretrained models, it can result in significant loss of that knowledge from catastrophic forgetting (French, 1999; Robins, 1995). Methods to mitigate this problem involve training only a part of the pretrained model, by linear probing, low-rank adaptation (Hu et al., 2021), and visual prompting (Bahng et al., 2022). However, these methods typically underperform on downstream tasks compared to fine-tuning the entire model.

Fine-tuning also reduces a model's robustness, which we take here to mean the ability to generalize to out-of-distribution (OOD) samples, as the model is optimized for a narrower distribution (Figure 1). Model robustness has been extensively studied by using various OOD datasets, typically beginning with an ImageNet pretrained model and evaluating it on OOD datasets that exhibit natural distribution shifts (Taori et al., 2020), such as changes in viewpoints (Barbu et al., 2019), time (Recht et al., 2019), styles (Hendrycks et al., 2021a; Wang et al., 2019), or synthetic data based on the original dataset (Hendrycks & Dietterich, 2019; Salvador & Oberman, 2022).

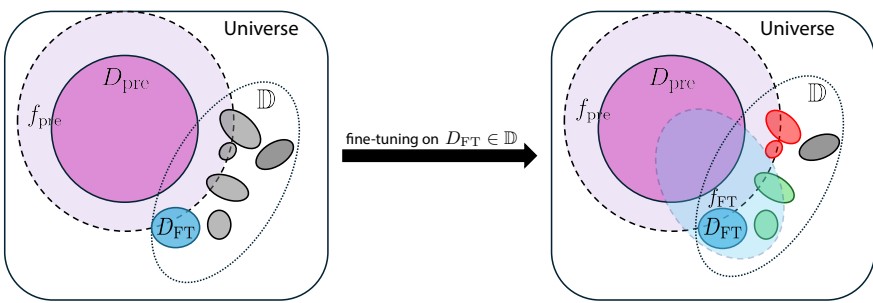

Figure 1: **Schematic: How fine-tuning changes the robustness of pretrained models.** A model pretrained on the dataset $D_{\mathrm{pre}}$ (purple solid) has a measure of robustness, generalizing to some out-of-distribution data (purple dashed, $f_{\mathrm{pre}}$). Dotted gray line: volume ($\mathbb{D}$) containing a number of related OOD datasets (dark gray ellipsoids). Fine-tuning on one of these datasets ($D_{\mathrm{FT}}$) shifts $f_{\mathrm{pre}}$ to $f_{\mathrm{FT}}$ (blue dashed ellipsoid), increasing performance on $D_{\mathrm{FT}}$ and some OOD tasks in $\mathbb{D}$ but possibly reducing performance on others (red), thus making the model less robust.

We observed that the ViT-B/16 CLIP (Radford et al., 2021) pretrained on LAION-2B suffers from more severe catastrophic forgetting on OOD datasets after fine-tuning on ImageNet-R (Hendrycks et al., 2021a) compared to the same model pretrained on ImageNet-21K (Ridnik et al., 2021) with AugReg (Steiner et al., 2022), despite their initially similar performance (Figure 2). Conversely, the ImageNet-21K pretrained model exhibits improved performance on ImageNet-Sketch (Wang et al., 2019).

To analyze why models pretrained on a smaller dataset have better OOD generalizability after fine-tuning, and the effect of the relationship between fine-tuning dataset and OOD datasets, we introduce ImageNet-RIB (Robustness Inheritance Benchmark), a new benchmark designed to assess the robustness of fine-tuned models across diverse downstream and evaluation OOD dataset pairs related to ImageNet. For each experiment, we fine-tune a pretrained model on a downstream dataset sampled from ImageNet OOD datasets and evaluate its performance on the remaining OOD datasets. This process is repeated across all available datasets to thoroughly assess how well the model retains robustness after fine-tuning. We also employ a variety of fine-tuning strategies, including vanilla fine-tuning, linear probing (fine-tuning the last layer only), LoRA (Hu et al., 2021), regularization-based continual learning methods (Li & Hoiem, 2017; Zenke et al., 2017), and robust fine-tuning methods (Kumar et al., 2022; Wortsman et al., 2022a;b).

Interestingly, pretraining on LAION-2B, despite its size and diversity, does not always yield the best results when fine-tuned on downstream datasets, suggesting that starting with large, rich

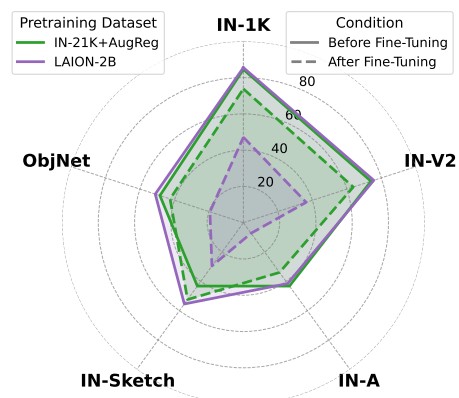

Figure 2: **OOD accuracy (robustness) of a ViT-B/16 model pretrained on two different datasets (LAION-2B, IN-21K with AugReg), before and after fine-tuning on ImageNet-R.**

datasets may not always be the optimal approach for preserving robustness, especially when the downstream dataset size is small. This problem occurs in the LAION-400M pretrained model, but not in the LAION-100M pretrained model. Our experimental results also show that the combination of regularization-based continual learning methods with model soup (Wortsman et al., 2022a) achieves the best performance in the benchmark, while linear probing performs the best when using LAION-2B pretrained models. Furthermore, our findings indicate that continual learning methods not only mitigate catastrophic forgetting related to the pretraining dataset but also enhance robustness when compared to standard fine-tuning. This improvement is attributed to leveraging the distributional properties of both pretraining and fine-tuning datasets.

In summary, the contributions of this paper are three-fold:

- We show that models pretrained on richer and larger datasets can have worse robustness after fine-tuning than models pretrained on smaller datasets if the fine-tuning dataset size is small. This degradation appears among the CLIP models and intensifies with pretraining scale, while the supervised models we test are largely unaffected (Section 4.3).

- We propose ImageNet-RIB, a new benchmark leveraging multiple ImageNet-based OOD datasets to quantify the robustness of fine-tuned models in comparison to pretrained models.

- We demonstrate that regularization-based continual learning methods improve robustness by leveraging both the pretraining and fine-tuning dataset distributions. This improvement is amplified when combined with robust fine-tuning methods.

## 2 Related Work

### 2.1 Robust Fine-Tuning Analysis

Robust fine-tuning aims to preserve the pretrained model's robustness to out-of-distribution (OOD) datasets, such as variations in viewpoint (Barbu et al., 2019), style (Hendrycks et al., 2021a; Wang et al., 2019), and temporal distribution shifts (Recht et al., 2019), during the fine-tuning. Large-scale transfer-learning studies (Wenzel et al., 2022; Andreassen et al., 2022) fine-tune thousands of ImageNet-pretrained models across many in-distribution (ID), OOD dataset pairs and show that ID and OOD accuracies typically co-vary, with truly "effectively robust" models being rare. Ramanujan et al. (2023) directly vary properties of the pre-training distribution and, on WILDS (Koh et al., 2021), find that increasing pre-training data quantity improves downstream robustness, whereas additional per-class diversity has a limited effect. We ask when robustness acquired during pretraining is inherited, degraded, or inverted by fine-tuning, and we find that models pretrained on the largest web-scale datasets (e.g., LAION-2B) can become less robust after fine-tuning than models pretrained on smaller curated datasets.

### 2.2 Robust Fine-Tuning Benchmark

Taori et al. (2020) proposed a benchmark that fine-tunes ImageNet-pretrained models on ImageNet-1K and evaluates robustness changes on multiple existing ImageNet-based OOD datasets. Though widely used (Kumar et al., 2022; Wortsman et al., 2022a;b), this setting fixes fine-tuning datasets and ignores pretraining dataset difference, which limits the analysis of how different downstream datasets or curricula affect robustness. Shi et al. (2023) extend this to joint training on two datasets (ImageNet-1K with CIFAR-10 (Krizhevsky et al., 2009) or YFCC (Thomee et al., 2016)), but still with a single pretraining distribution. Our benchmark is complementary to these studies. Rather than fixing the pretraining distribution and focusing on new algorithms, we vary the pretraining dataset (from small curated to large web-scale) while holding a family of related downstream datasets and OOD evaluation suites fixed. Please refer to Appendix A for comparison with single domain adaptation.

### 2.3 Robust Fine-Tuning Method

Wortsman et al. (2022a) demonstrate that averaging the parameters of multiple trained models improves both in-distribution and OOD performance. WiSE-FT (Wortsman et al., 2022b) further shows that linearly interpolating the weights of pretrained CLIP and ImageNet-1K fine-tuned CLIP improves robustness, although it requires tuning the interpolation ratio for optimal performance. LP-FT (Kumar et al., 2022) fine-tunes the last layer (linear probing) first and then fine-tunes the entire network. Goyal et al. (2023) show that contrastive learning using a text encoder in fine-tuning improves robustness. In this paper, we first adopt regularization-based continual-learning methods and combine them with existing method in robust fine-tuning.

### 2.4 Continual Learning

Continual learning aims to enable models to learn new tasks without forgetting previously learned knowledge. Existing approaches can be broadly categorized into three types: regularization-based methods, replay-based

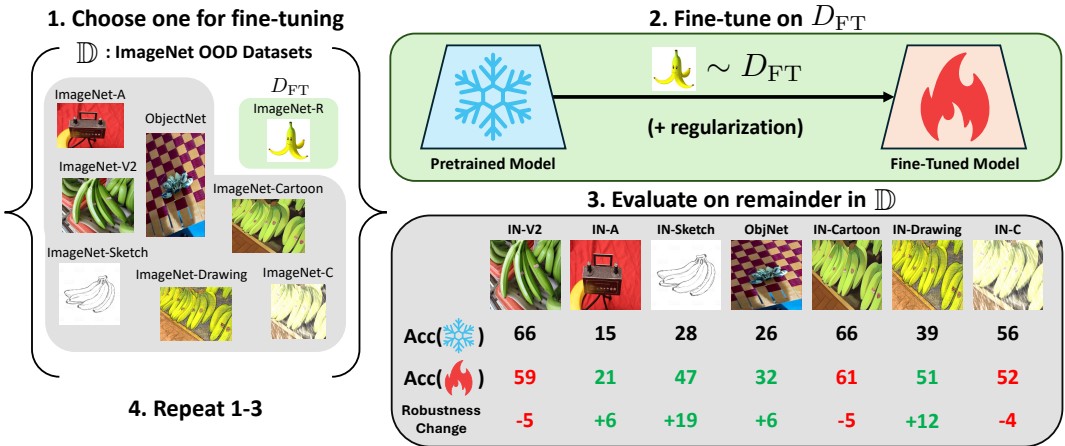

Figure 3: **ImageNet-RIB benchmarking process.** (1) We define a set $\mathbb{D}$ of ImageNet OOD datasets. We select one for fine-tuning, $D_{\mathrm{FT}}$, then assess the performance of the pretrained model on $\mathbb{D} \setminus D_{\mathrm{FT}}$. (2) After fine-tuning the pretrained model on $D_{\mathrm{FT}}$, we (3) re-assess its performance on $\mathbb{D} \setminus D_{\mathrm{FT}}$ and compute the robustness change. (4) This process is repeated until each dataset in $\mathbb{D}$ has been chosen once as the fine-tuning dataset, ensuring a detailed evaluation of fine-tuning's impact on robustness.

methods, and architecture-based methods. Regularization-based methods (Cheung et al., 2019; Kirkpatrick et al., 2017; Li & Hoiem, 2017; Zenke et al., 2017) use additional loss terms to limit changes to the model's parameters, ensuring that previously learned knowledge is retained. For instance, EWC (Kirkpatrick et al., 2017) employs the Fisher information matrix to determine the importance of each parameter, helping to preserve critical weights from earlier tasks. LwF (Li & Hoiem, 2017) uses knowledge distillation to transfer outputs from a model trained on past tasks to guide learning new tasks. Replay-based methods (Robins, 1995) mitigate catastrophic forgetting by creating a replay buffer that contains a subset of previous task data or synthetic data (Van de Ven et al., 2020) and a model is trained on the buffer along with a new task. Techniques such as reservoir sampling, reinforcement learning (Rebuffi et al., 2017), and gradient-based selection (Aljundi et al., 2019) help efficiently manage memory and select important data. Architecture-based methods modify the model's structure to accommodate new tasks by dynamically growing networks (Rusu et al., 2016; Wang et al., 2022; Yan et al., 2021). In our work, we focus on regularization-based continual learning methods to ensure a fair comparison with other fine-tuning approaches.

## 3 ImageNet Robustness Inheritance Benchmarking (ImageNet-RIB)

We propose the ImageNet-RIB (Robustness Inheritance Benchmark), a novel benchmark designed to measure robustness changes using existing ImageNet-related out-of-distribution (OOD) datasets as both fine-tuning and evaluation datasets. ImageNet-RIB fine-tunes pretrained models on various datasets, then evaluates robustness to other OOD datasets in the benchmark (Figure 3), offering a more comprehensive understanding of robustness fine-tuning.

### 3.1 Benchmark Protocol and Robustness Metric

**Protocol**  Figure 3 illustrates the protocol of our benchmark. Given a set of out-of-distribution (OOD) datasets $\mathbb{D} = \{D_1, D_2, ..., D_n\}$, we select one to use as a fine-tuning dataset $D_{\mathrm{FT}}$ for a pretrained model. We evaluate the model's performance on $\mathbb{D} \setminus D_{\mathrm{FT}}$ before and after fine-tuning on $D_{\mathrm{FT}}$, and compute the robustness change. This process is repeated by selecting each dataset in $\mathbb{D}$ as the fine-tuning dataset.

**Metric**  We define the robustness improvement score ($RI$) as the average relative robustness (Taori et al., 2020). Specifically, $RI$ measures the accuracy difference between fine-tuned and pretrained models on OOD

datasets. Formally, robustness improvement ($RI$) after fine-tuning on $D_i (= D_{\text{FT}})$ is defined as:

$$RI_i = \frac{1}{n-1} \sum_{j=1, j \neq i}^{n} A_i^{(j)} - A_{\text{pre}}^{(j)}, \tag{1}$$

where $A_{\text{pre}}^{(j)}$ and $A_i^{(j)}$ denote the average accuracies of pretrained and fine-tuned models on $D_j$, respectively. In addition to relative robustness, effective robustness (Taori et al., 2020) is an alternative metric commonly used to evaluate OOD performance (see Appendix E). Effective robustness measures how much the accuracy of a model deviates from an expected baseline, typically using a reference in-distribution dataset (*e.g.*, ImageNet-1K). While effective robustness is insightful, we use relative robustness in this benchmark to facilitate direct comparisons between different fine-tuning methods and initial pretraining datasets. We summarize the overall robustness improvement across all datasets as the mean robustness improvement ($mRI$).

### 3.2 Dataset Suites

We leverage all existing ImageNet OOD datasets to construct $\mathbb{D}$: ImageNet-V2 (Recht et al., 2019), ImageNet-A (Hendrycks et al., 2021b), ImageNet-Drawing (Salvador & Oberman, 2022), ImageNet-Cartoon (Salvador & Oberman, 2022), and ImageNet-Sketch (Wang et al., 2019), ObjectNet (Barbu et al., 2019), and ImageNet-C (Hendrycks & Dietterich, 2019). ObjectNet and ImageNet-C were originally designed solely for evaluating the OOD performance of ImageNet pretrained models, with restrictions on their use for training, however we extend their application in this benchmark by fine-tuning models on these datasets and evaluating their robustness on other OOD datasets. For detailed descriptions of each dataset, please refer to Appendix F.1. StanfordCars (Krause et al., 2013) dataset is also used as a showcase of a dataset with different label sets in Appendix 4.7.

## 4 Experiments

We use the ImageNet-RIB to assess the robustness of different pretrained models to fine-tune on a set of downstream datasets. The goal is to assess which fine-tuning methods do best across multiple pretraining datasets.

### 4.1 Experimental Details

**Pretrained Models** We use several architectures of Vision Transformer (ViT) (Dosovitskiy et al., 2021) and ResNet (He et al., 2016). The models are pretrained on ImageNet-1K (Russakovsky et al., 2015), or ImageNet-21K (Ridnik et al., 2021) and then fine-tuned on ImageNet-1K. The standard data augmentation and regularization technique for ViT, AugReg (Steiner et al., 2022) can be used for training on ImageNet-1K or ImageNet-21K. We also use ImageNet-1K with Sharpness Aware Minimization (SAM) (Chen et al., 2022), ImageNet-21K-P (Ridnik et al., 2021) pretrained models. Alternatively, some models are pretrained on LAION-2B (Schuhmann et al., 2022) or OpenAI internal dataset (400 million data) (Radford et al., 2021), followed by fine-tuning on ImageNet-1K. In other words, *all pretrained models are trained on ImageNet-1K before experiments* to directly leverage its classifier unless specified. For simplicity, we refer to them by the names of the first pretraining datasets (*e.g.*, LAION-2B). We also evaluate pretrained CLIP models with a zero-shot classifier that are not fine-tuned on ImageNet-1K in Appendix 4.4. In the main paper, we focus on ImageNet-1K with AugReg pretrained ViT-B/16 and experiments using other pretrained models are reported in Appendix H.

**Fine-tuning Methods** We employ standard fine-tuning methods, regularization-based continual learning methods for measuring performance on the proposed benchmark. The fine-tuning methods we evaluate include vanilla fine-tuning (FT), Linear Probing, LoRA (Hu et al., 2021), Visual Prompt (Bahng et al., 2022), LwF (Li & Hoiem, 2017), and EWC (Kirkpatrick et al., 2017)[1]. Because we are using ResNets, we do not

---

[1]We do not use other continual learning methods as the pretraining dataset is not accessible, and to ensure a fair comparison with other methods.

Table 1: Average robustness of the ViT-B/16 model pretrained on various datasets, assessed on the datasets in ImageNet-RIB set $\mathbb{D}$. LAION-2B pretraining exhibits the highest robustness.

| Pretraining Dataset | ImageNet-1K | IN-V2 | IN-A | IN-R | IN-Sketch | ObjNet | IN-Cartoon | IN-Drawing | IN-C |
|---|---|---|---|---|---|---|---|---|---|
| IN-1K + AugReg | 79.2 | 66.4 | 15.0 | 38.0 | 28.0 | 25.7 | 66.2 | 39.1 | 56.0 |
| IN-21K | 81.8 | 71.4 | 32.0 | 47.3 | 35.8 | 33.1 | 69.4 | 44.1 | 58.3 |
| IN-21K + AugReg | 84.5 | 74.0 | 43.2 | 56.8 | 43.2 | 39.1 | 75.1 | 54.9 | **66.5** |
| OpenAI | 85.3 | **75.7** | **47.3** | 65.9 | 50.9 | **50.7** | 76.3 | 55.7 | 62.6 |
| LAION-2B | **85.5** | 75.6 | 41.5 | **68.8** | **55.4** | 42.3 | **78.2** | **58.4** | 63.0 |

Table 2: *RI* and *mRI* of ViT-B/16 pretrained on ImageNet-1K and AugReg, fine-tuned on each of the datasets in the ImageNet-RIB set $\mathbb{D}$.

| Method | mRI | RI on specific $D_{\text{FT}}$ | | | | | | | |
|---|---|---|---|---|---|---|---|---|---|
| | | IN-V2 | IN-A | IN-R | IN-Sketch | ObjNet | IN-Cartoon | IN-Drawing | IN-C |
| FT | 1.3 | 2.9 | -4.0 | 2.8 | 4.4 | -2.7 | 0.6 | 0.4 | 5.9 |
| Linear Probing | 0.7 | 0.1 | -0.1 | 0.8 | 1.2 | 0.3 | 0.2 | 0.1 | 3.2 |
| Visual Prompt | -4.5 | -2.3 | -9.1 | -4.9 | -1.6 | -11.2 | -3.9 | -4.3 | 1.7 |
| LoRA | 0.9 | 0.2 | 0.4 | 1.1 | 2.6 | 0.3 | -0.1 | 1.3 | 1.1 |
| EWC | 2.8 | 2.9 | -0.2 | 5.2 | 4.4 | 1.4 | 1.6 | 2.8 | 4.3 |
| LwF | 3.1 | 2.8 | -0.0 | 6.2 | 4.6 | 0.7 | 1.9 | 2.1 | 6.5 |
| LP-FT | 2.3 | **3.0** | -0.9 | 5.2 | 4.5 | -0.1 | 1.2 | 0.6 | 4.7 |
| WiSE-FT | 3.6 | 2.5 | **0.7** | 7.5 | 4.5 | 2.1 | 2.3 | 3.0 | 6.5 |
| MS | **3.9** | 2.7 | 0.7 | **7.8** | **5.0** | **2.2** | **2.4** | **3.3** | **6.7** |

use LoRA, which was designed for ViT. We also employ robust fine-tuning methods: LP-FT (Kumar et al., 2022), WiSE-FT (Wortsman et al., 2022b), and uniform model soup (Wortsman et al., 2022a), which averages the parameters of a pretrained model, a vanilla fine-tuned model (FT), LwF, and EWC. We denote the uniform model soup, MS:PRE-FT-EWC-LwF to reveal the source of parameters. In Section 4.4, we also use FLYP (Goyal et al., 2023).

**Training**  Each pretrained model is fine-tuned on $D_{\text{FT}}$ for 10 epochs with a batch size of 64. We use stochastic gradient descent (SGD) with a learning rate of 0.001 and a momentum of 0.9 with cosine annealing (Loshchilov & Hutter, 2017). Visual Prompt is trained for 10 epochs with a learning rate of 40 without weight decay, following Bahng et al. (2022). We also evaluate models on ImageNet-RIB with a train-validation split of the fine-tuning dataset and select the best-performing models on the validation set for evaluation in Appendix D. Please refer to Appendix F.3 and the code repository for detailed implementation.

## 4.2 Combination of Continual Learning with Robust Fine-Tuning Methods Perform Best

**Baseline**  We start with the baseline of assessing model performance on the set of OOD datasets without any fine-tuning. Models pretrained on larger and more diverse datasets have better performance on both ImageNet-1K and downstream datasets as shown in Table 1. However, the ImageNet-21K with AugReg pretrained model achieves better performance on ImageNet-C than LAION-2B pretrained model since AugReg includes several corruptions in ImageNet-C (*e.g.*, brightness and contrast).

**Accuracy on OOD Datasets**  Table 30 presents the accuracy of an ImageNet-1K with AugReg pretrained ViT-B/16 model on OOD datasets before and after fine-tuning with each method on the fine-tuning dataset (see Table 31 for individual ImageNet-C corruption). Continual learning methods and robust fine-tuning methods generally improve performance on most OOD datasets after fine-tuning on the downstream datasets. Linear probing (LP) exhibits similar increase and decrease patterns as vanilla fine-tuning (FT), with less magnitude as the backbone network is fixed. Visual Prompt reduces performance even on ImageNet-1K after fine-tuning on synthetic datasets of the ImageNet validation set. This is inconsistent with Bahng et al. (2022), which showed its robustness to OOD datasets. This gap, however, reflects differences in the experimental setting. Bahng et al. (2022) learn a prompt on a frozen CLIP model with a zero-shot text classifier; in that setting, Visual Prompt is likewise among the most robust methods in our benchmark (Table 4). In contrast, with the fixed ImageNet-1K linear head used here, this single input-space perturbation fails to transfer and degrades accuracy, even on clean ImageNet-1K. A strong correlation exists between ImageNet-R, ImageNet-Sketch,

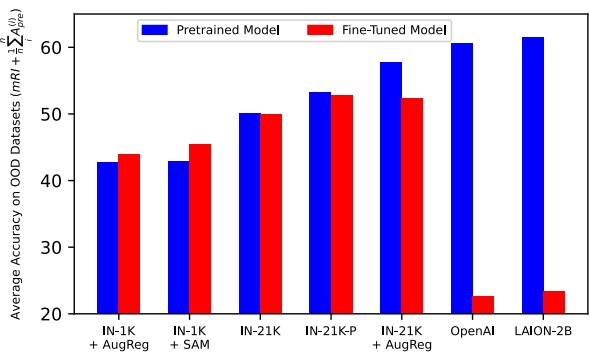

Figure 4: **Severe robustness loss from fine-tuning models pretrained on LAION-2B and OpenAI relative to fine-tuning models pretrained only smaller datasets.** The average accuracy on OOD datasets before (blue) and after (red) vanilla fine-tuning (see Figure 14 for other methods). The red bars are calculated by evaluating fine-tuned pretrained models on $\mathbb{D}$. The blue bars are the pretrained models' mean accuracy on $\mathbb{D}$. Fine-tuning models pretrained on LAION-2B and OpenAI causes severe robustness loss, leading to worse absolute performance on $\mathbb{D}$ than after fine-tuning an AugReg model pretrained on ImageNet-1K. Conversely, a model pretrained on ImageNet-1K with AugReg actually exhibits improved robustness after fine-tuning. Note that the difference between red and blue bars is $mRI$.

Table 3: Mean Robustness Improvement ($mRI$) after fine-tuning with different fine-tuning methods.

| Architecture → | ViT-B/16 | | | | | ViT-B/32 | | | | ViT-S/16 | | ViT-S/32 | ViT-L/16 | ResNet-18 | ResNet-50 |
|---|---|---|---|---|---|---|---|---|---|---|---|---|---|---|---|
| Method | IN-1K + AugReg | IN-21K | IN-21K + AugReg | OpenAI | LAION-2B | IN-1K + AugReg | IN-21K + AugReg | OpenAI | LAION-2B | IN-1K + AugReg | IN-21K + AugReg | IN-21K + AugReg | IN-21K + AugReg | IN-1K | IN-1K |
| FT | 1.3 | -0.1 | -5.5 | -38.0 | -38.1 | -0.0 | -0.1 | -28.7 | -31.6 | -3.2 | -2.3 | -2.9 | -2.1 | -5.2 | -5.2 |
| Linear Probing | 0.7 | 0.4 | -0.3 | **-2.0** | **-2.0** | 1.1 | 0.3 | **-1.3** | **-1.4** | 0.3 | -0.2 | -0.1 | -1.3 | -7.3 | -11.2 |
| Visual Prompt | -4.5 | -9.4 | -8.8 | -8.4 | -8.2 | -5.4 | -8.4 | -8.0 | -8.4 | -7.4 | -9.2 | -9.6 | -12.9 | -8.3 | -6.5 |
| LoRA | 0.9 | -0.3 | -2.1 | -3.6 | -3.6 | 0.9 | 0.9 | -1.8 | -1.9 | 0.9 | -1.5 | 0.4 | 1.0 | - | - |
| EWC | 2.8 | 1.4 | 0.6 | -12.7 | -12.5 | 1.3 | 1.6 | -7.0 | -10.0 | 1.6 | 1.6 | 1.0 | 1.1 | -5.7 | -8.9 |
| LwF | 3.1 | 1.6 | -1.0 | -33.1 | -33.9 | 1.8 | 1.7 | -23.9 | -26.7 | 0.6 | 0.5 | 0.3 | -0.2 | -1.9 | -5.8 |
| LP-FT | 2.3 | 0.5 | -2.6 | -36.9 | -37.1 | 1.5 | 1.2 | -27.7 | -30.8 | -1.2 | -0.8 | -1.1 | -3.5 | -4.8 | -5.1 |
| WiSE-FT | 3.6 | 2.5 | 1.7 | -18.1 | -21.6 | 2.5 | **3.0** | -9.7 | -13.5 | 2.9 | 2.8 | 2.3 | 2.3 | **0.7** | **1.2** |
| MS | **3.9** | **2.7** | **2.2** | -16.0 | -17.9 | **2.5** | 2.8 | -8.1 | -10.9 | **3.0** | **2.3** | **2.8** | **2.5** | -0.1 | -0.5 |

and ImageNet-Drawing, as they share drawing and sketch renditions, and ImageNet-R and ImageNet-Sketch share images. Fine-tuning on ImageNet-C improves performance on other synthetic datasets, but not vice versa, due to its diverse corruptions and severities.

**Robustness Improvement** Individual robustness improvement scores ($RI$) after fine-tuning on each OOD dataset with ImageNet-1K with AugReg pretrained ViT-B/16 also show that MS:PRE-FT-EWC-LwF consistently performs the highest in most datasets, followed by WiSE-FT as demonstrated in Table 2. This is because they directly use the weights of pretrained models, thus taking advantage of their robustness.

**Mean Robustness Improvement** The combination of continual learning methods with weight averaging (MS:PRE-FT-EWC-LwF) achieves the highest or second-highest mean robustness improvement ($mRI$) across different backbones pretrained on ImageNet-based datasets as shown in Table 3. Moreover, end-to-end continual learning methods show comparable performance to the multi-stage method (Kumar et al., 2022) or the post-hoc robustness method (Wortsman et al., 2022b). This shows the potential of continual learning methods in the field of robust fine-tuning. The robustness of linear probing and Visual Prompt remains relatively unchanged since they do not modify the models' weights significantly but their performance on the downstream dataset tends to be worse (see Appendix H.3). Consequently, they have much better performance with LAION-2B pretrained models compared to other methods, which show a significant robustness decrease.

## 4.3 Paradoxically, Models pretrained on the Largest Datasets Do Worst After Fine-Tuning

The extent of robustness degradation increases with the size and diversity of the pretraining dataset, as illustrated in Table 3 and Figure 4. As a result, the robustness of fine-tuned models pretrained on larger datasets (*e.g.*, LAION-2B, OpenAI) exhibits worse robustness compared to those pretrained on smaller

Table 4: $mRI$ of ViT-B/16, ViT-B/32, ConvNeXt-Base, and ResNet-50 CLIP models, and SigLip-B/16, SigLip2-B/16 pretrained on various datasets using zero-shot classifier (Radford et al., 2021).

| Method | ViT-B/16 | | | ViT-B/32 | | | | ConvNeXt | ResNet-50 | | | SigLip | SigLip2 |
| | LAION 400M | OpenAI | LAION 2B | LAION 100M | LAION 400M | OpenAI | LAION 2B | LAION-2B | CC 12M | YFCC 15M | OpenAI | WebLI | WebLI |
|---|---|---|---|---|---|---|---|---|---|---|---|---|---|
| FT | -32.5 | -31.2 | -37.4 | -7.3 | -23.8 | -27.0 | -33.0 | -16.9 | -7.1 | -2.6 | -24.1 | -21.3 | -25.7 |
| FYLP | -26.0 | -30.9 | -36.6 | -4.6 | -24.5 | -26.9 | -32.8 | -17.5 | **-4.4** | **-0.7** | -26.1 | -24.0 | -28.7 |
| Visual Prompt | **-7.6** | -11.0 | **-10.2** | -11.6 | -9.6 | -7.7 | **-10.4** | -53.5 | -8.9 | -6.3 | **-10.9** | -26.7 | -25.6 |
| LoRA | -47.8 | -49.2 | -52.5 | -35.2 | -41.3 | -41.1 | -19.4 | - | - | - | - | -47.4 | -20.7 |
| EWC | -8.1 | **-8.8** | -13.1 | -1.5 | **-5.4** | **-7.3** | -10.5 | -5.5 | -8.8 | -5.8 | -17.2 | **-1.9** | -3.1 |
| LwF | -29.6 | -24.9 | -30.9 | -2.1 | -16.8 | -22.5 | -29.1 | -11.8 | -6.2 | -1.7 | -22.1 | -12.9 | -17.2 |
| WiSE-FT | -19.5 | -14.4 | -23.2 | 1.7 | -9.5 | -11.9 | -17.5 | -4.3 | -8.3 | -3.4 | -29.1 | -4.3 | -7.3 |
| MS | -20.4 | -11.3 | -18.9 | **2.3** | -15.0 | -9.4 | -15.0 | **-3.9** | -10.3 | -3.5 | -35.1 | -2.2 | **-4.2** |

Table 5: Average zero-shot accuracy on ImageNet-1K validation set and each OOD dataset using pretrained ViT-B/16 and ResNet-50 CLIP models. All pretrained weights are acquired from open_clip library.

| Architecture | Pretraining Dataset | IN-1K | Avg. OOD | IN-V2 | IN-A | IN-R | IN-Sketch | ObjNet | IN-Cartoon | IN-Drawing | IN-C |
|---|---|---|---|---|---|---|---|---|---|---|---|
| | LAION-400M | 67.1 | 50.2 | 59.6 | 33.1 | 77.9 | 52.4 | 46.1 | 56.1 | 36.4 | 40.2 |
| ViT-B/16 | OpenAI | 64.4 | 48.8 | 57.8 | 44.4 | 73.5 | 44.3 | 50.5 | 48.2 | 33.3 | 38.2 |
| | LAION-2B | 70.2 | 53.5 | 62.3 | 38.0 | 80.6 | 56.1 | 50.8 | 59.2 | 37.5 | 43.0 |
| | LAION-100M | 52.5 | 38.6 | 44.5 | 14.6 | 64.5 | 39.8 | 30.0 | 43.9 | 44.5 | 27.3 |
| ViT-B/32 | LAION-400M | 60.2 | 42.6 | 52.4 | 19.6 | 70.8 | 46.4 | 38.9 | 48.4 | 29.4 | 35.1 |
| | OpenAI | 59.6 | 41.4 | 52.9 | 28.3 | 67.1 | 40.4 | 31.6 | 46.1 | 29.1 | 35.9 |
| | LAION-2B | 66.6 | 48.7 | 58.1 | 26.3 | 76.4 | 53.7 | 44.7 | 55.8 | 33.2 | 41.6 |
| ConvNeXT | LAION-2B | 70.8 | 57.2 | 62.4 | 40.1 | 80.7 | 57.1 | 52.2 | 58.5 | 62.4 | 44.5 |
| | CC-12M | 35.9 | 21.4 | 30.6 | 7.5 | 44.6 | 23.5 | 21.8 | 23.8 | 9.1 | 10.1 |
| ResNet-50 | YFCC-15M | 32.3 | 15.3 | 28.0 | 13.7 | 22.2 | 7.3 | 16.7 | 17.4 | 6.8 | 10.0 |
| | OpenAI | 57.9 | 36.8 | 50.9 | 23.3 | 60.1 | 34.6 | 36.9 | 40.2 | 22.4 | 25.6 |
| SigLIP | WebLI | 76.1 | 60.6 | 69.0 | 45.0 | 90.2 | 67.9 | 50.8 | 67.4 | 47.5 | 47.1 |
| SigLip2 | WebLI | 69.7 | 59.3 | 63.4 | 54.1 | 83.9 | 61.8 | 53.8 | 63.3 | 47.0 | 47.2 |

datasets and their corresponding fine-tuned counterparts when using vanilla fine-tuning. Similarly, the LAION-400M CLIP model has better robustness than the LAION-2B model with a zero-shot classifier from a text encoder after fine-tuning (see Table 6). As shown in Appendix D, the discrepancy between the LAION-2B and ImageNet-1K + AugReg pretrained models increases when selecting the best-performing epoch for each case instead of using a fixed 10-epoch duration.

One possible explanation is that models pretrained on larger, more diverse datasets have more room for performance degradation from catastrophic forgetting as they demonstrate higher initial robustness (see Table 1). However, this does not fully explain the pronounced robustness loss observed in OpenAI or LAION-2B pretrained models, particularly when compared to ImageNet-21K with AugReg pretrained models, which exhibit similar initial robustness. Notably, ImageNet-21K and its variants begin to exhibit robustness degradation, especially when using vanilla fine-tuning. This could be an early indicator of performance decay in larger pretrained models. Although ImageNet-21K is the second-largest dataset with 14 million images, it is much smaller than LAION-2B, which contains two billion images. We hypothesize that this discrepancy in pretraining dataset size contributes to the difference in robustness degradation.

## 4.4 CLIP with Zero-Shot Classifier on ImageNet-RIB

In Section 4.2, we evaluated models pretrained on various datasets and subsequently fine-tuned on ImageNet-1K for the ImageNet-RIB benchmark. Here, we extend this analysis to measure the $mRI$ of CLIP models that bypass ImageNet-1K fine-tuning and are directly fine-tuned on downstream datasets. We utilize pretrained weights from the open_clip library (Ilharco et al., 2021) and adopt a zero-shot classifier, as proposed by Radford et al. (2021), instead of a linear readout layer. This choice is driven by the fact that each OOD dataset contains a distinct subset of ImageNet-1K labels, making a unified linear probe impractical. For example, fine-tuning on ImageNet-A involves only 200 classes, whereas ImageNet-Sketch covers 1,000 classes. Consequently, methods such as Linear Probing and LP-FT are excluded, as zero-shot classifiers do not require additional training.

Table 6: Average Accuracy on OOD Datasets of each ViT-B/16 CLIP model with zero-shot classifier before and after vanilla fine-tuning (FT). LAION-400M pretrained model outperforms LAION-2B pretrained one after fine-tuning.

| ViT-B/16 CLIP | OpenAI | LAION-400M | LAION-2B |
|---|---|---|---|
| Before Fine-Tuned | 48.8 | 50.2 | **53.5** |
| After Fine-Tuned | 16.3 | **19.0** | 16.1 |

Table 7: $mRI$ of ViT-B/32 CLIP with zero-shot classifier after vanilla fine-tuning (FT). Only OpenAI pretrained model has huge negative $mRI$.

| ViT-B/32 | LAION-100M | LAION-400M | LAION-2B |
|---|---|---|---|
| $mRI$ | **-7.3** | -23.8 | -33.0 |

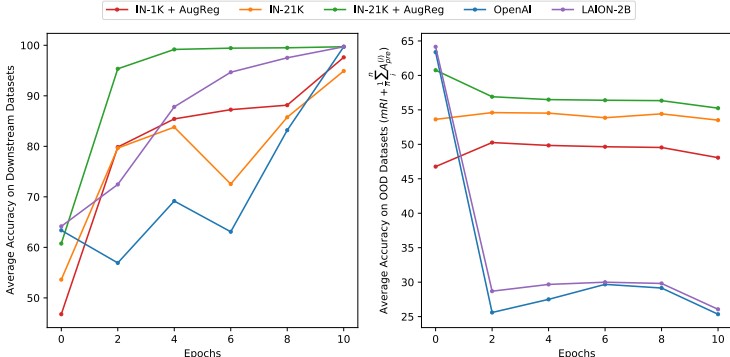

Figure 5: **Fine-tuning LAION-2B and OpenAI pretrained models causes severe robustness loss while learning slower than ImageNet-21K pretrained AugReg model.** The average accuracy on fine-tuning datasets (left) and the average accuracy on OOD datasets (right) while fine-tuning on the downstream dataset using vanilla fine-tuning method (FT) with ViT-B/16. Although these models learn slower than other methods, they suffer from a huge robustness drop even in the early period of fine-tuning.

Table 4 reports the $mRI$ of various ViT-B/16, ViT-B/32, ConvNeXt-Base (Liu et al., 2022) and ResNet-50 CLIP models and SigLip-B/16 (Zhai et al., 2023), SigLip2-B/16 (Tschannen et al., 2025) pretrained on different datasets without fine-tuned on ImageNet-1K. FLYP (Goyal et al., 2023) is effective on ResNet-50 CLIP pretrained on a smaller dataset, while it does not solve a problem in other models. Consistent with the results in Table 3, we observe that most fine-tuning approaches, including vanilla fine-tuning, significantly degrade robustness when using models pretrained on the large-scale dataset. However, ViT-B/32 CLIP model pretrained on LAION-100M (Lin et al., 2024) and ResNet-50 CLIP models pretrained on Conceptual-12M (CC-12M) (Changpinyo et al., 2021) and YFCC-15M (Thomee et al., 2016) do not exhibit this robustness degradation. This suggests that a CLIP model pretrained on comparatively smaller datasets experiences less catastrophic forgetting in out-of-distribution generalization than a model pretrained on larger datasets. Additionally, Table 5 presents the zero-shot accuracy of pretrained CLIP models on the ImageNet-1K validation set and each OOD dataset. Taking into account both the average OOD accuracy and $mRI$, LAION-400M outperforms LAION-2B, achieving a 2.9-point higher OOD accuracy after vanilla fine-tuning (19.0 vs. 16.1).

## 4.5 Analysis of Severe Catastrophic Forgetting in Models Pretrained on Large Datasets

**CLIP pretrained on Small Dataset Does Not Exhibit Severe Robustness Degradation.** To discern whether the robustness degradation arises from the CLIP pretraining method itself or from the scale of the pretraining data, we evaluate CLIP models trained on smaller datasets. The ViT-B/32 CLIP model pretrained on LAION-100M shows much less degradation, whereas counterparts trained on LAION-400M and LAION-2B do (see Table 7) although they have similar performance before fine-tuning as shown in Table 5. Similarly, ResNet-50 CLIP models pretrained on CC-12M (Changpinyo et al., 2021) and YFCC-15M (Thomee et al., 2016) remain robust, while the model pretrained on OpenAI's internal 400M-image dataset exhibits pronounced degradation as we mentioned in Section 4.4.

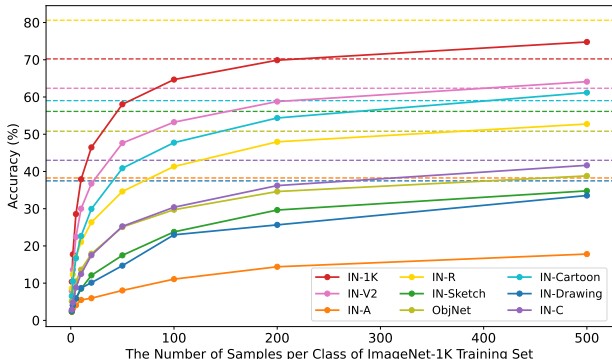

Figure 6: **Fine-tuning on small dataset leads to severe accuracy degradation both in- and out-of-distribution.** Accuracy on ImageNet-1K validation set and OOD datasets after fine-tuning LAION-2B pretrained ViT-B/16 CLIP model with zero-shot classifier on a small number of images per class of ImageNet-1K training set. The dashed line denotes the accuracy of the pretrained model on each dataset.

Table 8: Accuracy on each dataset of ViT-B/16 pretrained models after fine-tuning on ImageNet-1K validation set. The parenthesis denotes the difference with pretrained models.

| Pretraining Dataset | IN-1K | IN-V2 | IN-A | IN-R | IN-Sketch | ObjNet | IN-Cartoon | IN-Drawing | IN-C |
|---|---|---|---|---|---|---|---|---|---|
| IN-1K + AugReg | 97.5 (+18.3) | 66.9 (+0.4) | 23.3 (+8.3) | 40.9 (+2.9) | 29.5 (+1.5) | 37.2 (+4.2) | 71.1 (+4.9) | 41.0 (+1.9) | 59.5 (+3.5) |
| IN-1K + SAM | 87.3 (+7.1) | 69.4 (+1.2) | 17.7 (+8.7) | 41.8 (+1.7) | 30.1 (+2.4) | 38 (+3.8) | 72.1 (+5.2) | 42.9 (+0.6) | 56.9 (+2.3) |
| IN-21K | 94.7 (+12.9) | 71.6 (+0.2) | 38.5 (+6.5) | 49.9 (+2.6) | 36.7 (+0.9) | 45.2 (+2.7) | 73.9 (+4.5) | 44.1 (0.0) | 59.8 (+1.5) |
| IN-21K-P | 96.9 (+12.6) | 73.0 (-1.0) | 41.4 (+7.3) | 51.5 (0.0) | 39.8 (-0.4) | 45.8 (-0.9) | 76.4 (+2.9) | 44.3 (-0.8) | 61.7 (+0.3) |
| IN-21K + AugReg | 99.9 (+15.4) | 70.6 (-3.4) | 42.2 (-1.0) | 54.1 (-2.7) | 39.4 (-3.8) | 47.9 (-0.5) | 84.5 (+9.4) | 55.5 (+0.6) | 69.7 (+3.2) |
| OpenAI | 99.9 (+14.6) | 59.9 (-15.8) | 13.9 (-33.4) | 34.9 (-31.0) | 19.7 (-31.2) | 30.5 (-20.2) | 75.0 (-1.3) | 33.4 (-22.3) | 45.7 (-16.9) |
| LAION-2B | 99.9 (+14.4) | 59.4 (-16.2) | 12.6 (-28.9) | 36.3 (-32.5) | 23.4 (-32.0) | 30.4 (-20.7) | 73.0 (-5.2) | 30.6 (-27.8) | 41.8 (-21.2) |

**Overfitting Does Not Drive Robustness Collapse.** A plausible explanation for the observed decline in robustness during fine-tuning could be early overfitting in LAION-2B and OpenAI pretrained ViT-B/16 models. We investigate this by tracking robustness performance and average accuracy on the downstream datasets throughout standard fine-tuning (FT). Figure 5 reveals that the ImageNet-21K model pretrained with AugReg learns the fine-tuning dataset faster than other methods, while OpenAI pretrained model shows the slowest learning progression. Despite this, only the LAION-2B and OpenAI models experience significant OOD robustness degradation. This finding indicates that overfitting is not the primary driver of catastrophic forgetting in these models. Moreover, Appendix G.1 shows that catastrophic forgetting occurs even with a much smaller learning rate.

**Fine-Tuning Dataset Texture Does Not Account for Forgetting.** Unlike traditional benchmarks (Taori et al., 2020), which use natural images (*e.g.*, ImageNet-1K), the ImageNet-RIB benchmark incorporates a variety of styles, including cartoons, drawings, and sketches. One may hypothesize that models pretrained on large datasets are susceptible to robustness degradation when fine-tuned on downstream datasets featuring stylized or non-natural images. However, our findings challenge this hypothesis; fine-tuning on the ImageNet-1K validation set also leads to similar robustness collapse, even though *all* models are pretrained and then fine-tuned on ImageNet-1K training set (see Table 8).

**Fine-Tuning Dataset Size is a Major Determinant.** The consistent robustness degradation seen in OpenAI and LAION-2B pretrained models fine-tuned on the ImageNet-1K validation set leads us to hypothesize that the size of the downstream dataset plays a significant role in catastrophic forgetting. While Ramanujan et al. (2023) and Fang et al. (2022) demonstrate that CLIP's robustness is primarily attributed to the pretraining dataset size and distribution—rather than contrastive learning—the impact of downstream dataset size remains underexplored. To investigate, we fine-tune a LAION-2B pretrained CLIP model (not previously fine-tuned on ImageNet-1K) on subsets of the ImageNet-1K training set, using a zero-shot classifier similar to Section 4.4. As shown in Figure 6, both ImageNet-1K validation accuracy and OOD performance

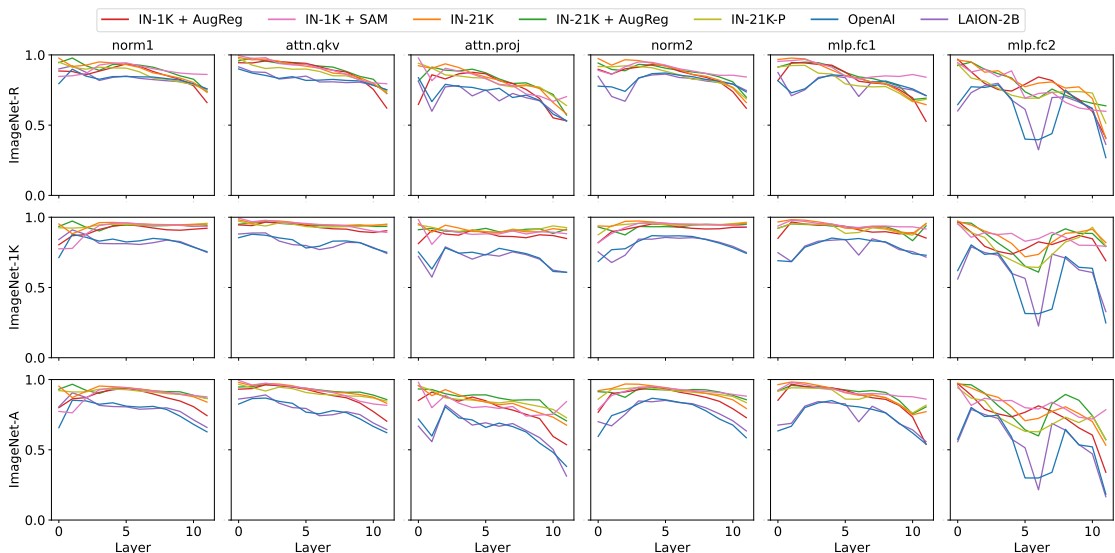

Figure 7: **Representational shifts from fine-tuning on ImageNet-R, analyzed by Centered Kernel Alignment (CKA), in ViT-B/16 models pretrained on various datasets.** Rows indicate datasets where CKA is measured and column indicates different part of transformer demonstrated in Figure 9. ImageNet-R (downstream), ImageNet-1K (pretraining), and ImageNet-A (OOD dataset) are used as evaluation datasets. Models pretrained on OpenAI and LAION-2B show distinct CKA patterns across layers compared to others.

Table 9: Average accuracy on ImageNet-1K validation set and each OOD dataset using ViT-B/32 CLIP with zero-shot classifier before and after fine-tuning on the StanfordCars Dataset. Bold and denotes the best performance of pretrained and fine-tuned models, respectively.

| Pretraining Dataset | ImageNet-1K | AvgOOD | IN-V2 | IN-A | IN-R | IN-Sketch | ObjNet | IN-Cartoon | IN-Drawing | IN-C |
|---|---|---|---|---|---|---|---|---|---|---|
| LAION-100M (Pretrained) | 52.5 | 38.6 | 44.5 | 14.6 | 64.5 | 39.8 | 30 | 43.9 | **44.5** | 27.3 |
| LAION-100M (Fine-tuned) | *43.9* | *28.6* | ***36.2*** | *8.7* | *53.9* | *29.7* | *23.2* | *34.1* | *22.0* | *20.8* |
| LAION-400M (Pretrained) | 59.6 | 42.6 | 52.4 | 19.6 | 70.8 | 46.4 | 38.9 | 48.4 | 29.4 | 35.1 |
| LAION-400M (Fine-tuned) | 14.2 | 9.2 | 12.5 | 4.0 | 19.6 | 7.3 | 9.3 | 9.8 | 5.3 | 5.8 |
| OpenAI (Pretrained) | 60.2 | 41.4 | 52.9 | 28.3 | 67.1 | 40.4 | 31.6 | 46.1 | 29.1 | 35.9 |
| OpenAI (Fine-tuned) | 5.7 | 3.9 | 4.6 | 2.1 | 8.7 | 2.3 | 5.1 | 3.9 | 2.1 | 2.6 |
| LAION-2B (Pretrained) | **66.6** | **48.7** | **58.1** | **26.3** | **76.4** | **53.7** | **44.7** | **55.8** | 33.2 | **41.6** |
| LAION-2B (Fine-tuned) | 6.8 | 4.8 | 5.8 | 2.1 | 10.7 | 3.7 | 5.7 | 5.0 | 2.5 | 2.9 |

degrade significantly when fine-tuned on smaller subsets. This degradation persists across hyperparameter variations, including learning rate and training epochs. These findings indicate that CLIP models require sufficiently large fine-tuning datasets to maintain robustness against distribution shifts. Insufficient data in fine-tuning likely exacerbates catastrophic forgetting, highlighting dataset size as a critical factor in mitigating OOD performance decline.

## 4.6 Analysis of Representation Shift via Centered Kernel Alignment (CKA)

To analyze intermediate representations before and after fine-tuning, we use Centered Kernel Alignment (CKA) (Cortes et al., 2012; Kornblith et al., 2019), the standard metrics to quantify similarity between neural network representations (Kim & Han, 2023; Raghu et al., 2021). We fine-tune ViT-B/16 models pretrained on various datasets on ImageNet-R and measure the CKA between the pretrained and fine-tuned models on ImageNet-R (fine-tuning dataset), ImageNet-1K validation set, and ImageNet-A (OOD dataset). Figure 7 shows the CKA scores across transformer layers, broken down by component, as illustrated in Figure 9. Deeper layers exhibit greater discrepancy between the pretrained and fine-tuned models. Models pretrained on LAION-2B and OpenAI's internal dataset show especially large discrepancies, particularly in earlier layers,

compared to other models. In addition, we observe a pronounced change in the mlp.fc2 component of the sixth transformer block, in which pattern was observed in Li et al. (2024). This change becomes more significant in models pretrained on larger datasets, especially when evaluated on non-downstream datasets. Considering the fact that this huge dissimilarity is not observed in the previous layer (mlp.fc1). We expect that the sixth mlp.fc2 layer may be linked to severe catastrophic forgetting, warranting further investigation. This trend is maintained across various combinations of downstream datasets as shown in Figures S1-S8 in the supplementary materials.

Freezing this layer during fine-tuning, however, does not prevent the collapse. Specifically, for the LAION-2B pretrained ViT-B/16, $mRI$ does not improve but instead worsens slightly, dropping from $-41.7$ under full fine-tuning to $-50.1$. This indicates that the shift at the sixth mlp.fc2 is a correlate of the collapse rather than a localized cause that freezing can prevent, leaving its mechanism open for future work.

### 4.7   Stanford Car Dataset

We fine-tune various pretrained ViT-B/32 CLIPs on the StanfordCars dataset (Krause et al., 2013) with a zero-shot classifier (FT) and then evaluate their robustness on the full suite of OOD datasets used in the paper. The dataset has 196 fine-grained car classes, unlike ImageNet variants. As shown in Table 9, the results align with our findings that a model pretrained on a larger dataset suffers more catastrophic forgetting. The LAION-100M pretrained model does not suffer from severe performance degradation, whereas the other models do. This supports our hypothesis that pretraining on a large-scale dataset (LAION-400M, OpenAI, LAION-2B) is more likely to lead to severe catastrophic forgetting than pretraining on a smaller dataset (LAION-100M).

## 5   Discussion

We found that models pretrained on larger, more diverse datasets, such as LAION-2B, experienced more severe robustness degradation after fine-tuning While these models exhibited high initial robustness, the performance drop was more prominent compared to models pretrained on smaller datasets like ImageNet-1K or LAION-100M, leading to even worse performance. This degradation arises in the CLIP models and grows as their pretraining data scales, whereas the supervised models we evaluate do not show it; we scope our claim accordingly (Section 4.3). To facilitate these analyses, we introduced ImageNet-RIB (Robustness Inheritance Benchmark), a framework that evaluates model robustness across multiple downstream and OOD dataset pairs. In contrast to existing benchmarks that primarily consider a single downstream dataset (Taori et al., 2020), ImageNet-RIB enables nuanced analyses of how varying fine-tuning contexts influence model generalization. Moreover, we demonstrated that continual learning methods and robust fine-tuning approaches, particularly in combination, are effective in preserving or even improving robustness. Specifically, the combination of model soup with continual learning techniques consistently achieved superior performance. This finding underscores the potential of integrating these strategies to mitigate catastrophic forgetting and enhance the robustness to OOD datasets.

Despite these contributions, our study has limitations. Although we identify conditions under which extensive pretraining negatively impacts robustness and analyze feature representation, the underlying mechanisms remain unclear. In particular, freezing the layer with the largest representational shift (the sixth mlp.fc2) during fine-tuning does not prevent the collapse (Section 4.6), suggesting the shift is a correlate rather than a localized cause. Future research should investigate why extensive pretraining leads to worse robustness compared to smaller-scale pretraining, potentially informing more effective fine-tuning strategies. Extending our evaluation to additional architectures and dataset contexts would further strengthen and generalize our conclusions. In summary, our findings challenge common assumptions about pretraining dataset scale and robustness, emphasizing the importance of tailored fine-tuning strategies. We hope these insights motivate further investigation into optimizing robust fine-tuning practices, ultimately advancing the reliability and generalization capabilities of machine learning models.

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

## Appendix

We summarize each section in the Appendix as follows:

**Appendix A (Additional Related Works - Single Domain Generalization)**   We survey single domain generalization

**Appendix B (Relationship between Dataset Distance and Accuracy Drop on Pretraining Dataset).**   We quantify how distributional distance from ImageNet-1K relates to post-tuning performance.

**Appendix C (Fine-Tuning on Small Subset of Fine-Tuning Dataset).**   We show that the robustness degradation happens when the dataset size is small.

**Appendix D (ImageNet-RIB with Train-Validation Split).**   We split the fine-tuning dataset into a train set and a validation set and find the best-performing model on each setting.

**Appendix E (Effective Robustness).**   We demonstrate $mRI$ with effective robustness from the results in Appendix D.

**Appendix F (Experimental Details).**   We describe experimental details.

**Appendix G (Ablation Studies).**   We conduct various ablation studies including learning rate, weight decay, multiple random seeds, and best ratio for WiSE-FT.

**Appendix H (Additional Experiments with Various Pretrained Models).**   We include extensive experimental results such as robustness of pretrained models, backward transfer, performance on fine-tuning dataset, and full accuracies on experimental settings.

## A   Additional Related Works: Single Domain Generalization

Single-domain generalization refers to the task where only one source domain is available during training, and the model is evaluated on multiple unseen target domains (Qiao et al., 2020). While the high-level concept is similar to the existing robust fine-tuning benchmark (Taori et al., 2020), the objectives differ. Robust fine-tuning focuses on maintaining or improving a model's robustness to OOD datasets during fine-tuning, whereas single-domain generalization aims to achieve generalization to unseen OOD datasets, often through meta-learning-based data augmentation (Chen et al., 2023; Qiao et al., 2020) or adaptive batch normalization (Fan et al., 2021). Recently, Fan et al. (2021) apply single-domain generalization to the PACS dataset (Li et al., 2017), using one domain as the training set and the remaining domains as test sets. This setup resembles our ImageNet-RIB benchmark in that each dataset is used for training while the others are used for testing. However, the goals of the two benchmarks differ: our robust fine-tuning benchmark aims to mitigate robustness degradation during fine-tuning, while single-domain generalization benchmarks focus on improving generalizability from a single source domain.

## B   Relationship between Dataset Distance and Accuracy Drop on Pretraining Dataset

### B.1   Optimal Transport Dataset Distance on Feature Space Aligns with Dataset Design Principles

We measure the distance between datasets by using Optimal Transport Dataset Distance (OTDD) (Alvarez-Melis & Fusi, 2020) and Normalized Compression Distance (NCD) (Cilibrasi & Vitányi, 2005). The distance is measured in the image space and the feature space from ImageNet-1K with AugReg pretrained ViT-B/16, class tokens before the classifier layer. Since ImageNet-C comprises multiple corruptions with different severities, we do not measure the distance to ImageNet-C. OTDD in the image space, ImageNet-Sketch is the farthest from other datasets as it is black and white sketch images (Figure 10a). ImageNet-Drawing is the

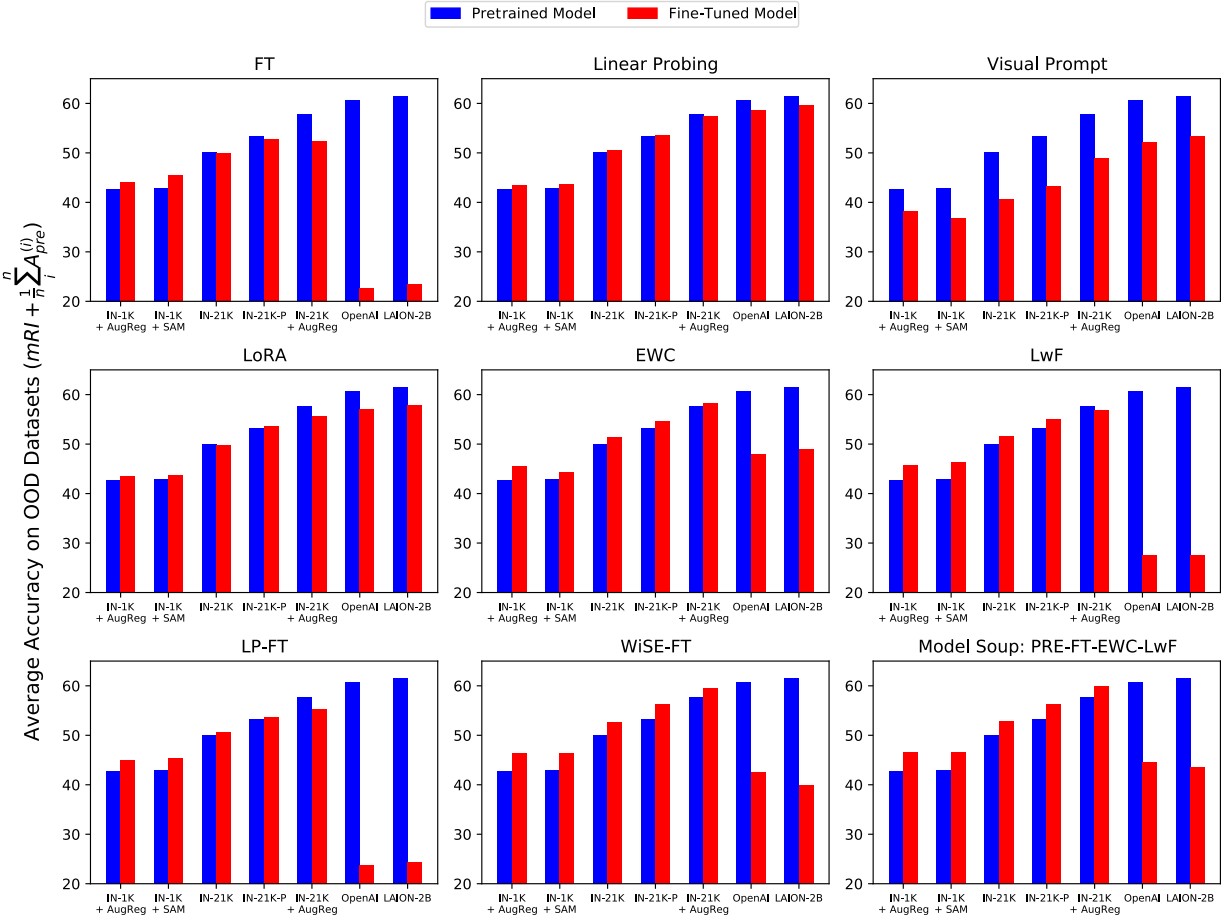

Figure 8: **The average accuracy on OOD datasets before (blue) and after (red) fine-tuning with each method on fine-tuning datasets.** The red bar is calculated directly by evaluating pretrained models on OOD datasets while the blue bar is calculated by adding $mRI$ of each method to the pretrained models' accuracy. Note that it is identical to the average accuracy on OOD datasets after fine-tuning on each dataset ($mRI + \frac{1}{n}\sum_i^n A_{\text{pre}}^{(i)} = \frac{1}{n}\sum_j \frac{1}{n-1}\sum_{i,i\neq j}^n A_{\text{down}}^{(i)}$). Fine-tuning LAION-2B and OpenAI pretrained models on the fine-tuning datasets causes severe robustness loss leading to worse performance than ImageNet-1K with AugReg pretrained model. Conversely, ImageNet-1K with the AugReg pretrained model improves robustness after fine-tuning. Note that the difference between red and blue bars is $mRI$.

closest to the dataset and the ImageNet-R is the second closest as they share the same styles and images, respectively.

OTDD in the feature space demonstrates a better alignment with the dataset design principles (Figure 10b). For example, ImageNet-V2 is designed to replicate the distribution of the ImageNet validation set. It leads ImageNet-V2 the closest to ImageNet-1K among realistic datasets. Moreover, the distances between ImageNet-1K and ImageNet-V2 to other datasets are consistent across both image and feature spaces. This is not true with ImageNet-Cartoon since it is a synthetic dataset based on the ImageNet validation set. As shown in Table 30, ImageNet-Cartoon improves ImageNet-1K accuracy more than ImageNet-Drawing, suggesting that the distribution shift in cartoon-style images is less severe than that of drawing-style images. Similarly, ObjectNet is intentionally collected with different viewpoints and backgrounds and it is the most distant from all other datasets in the feature space.

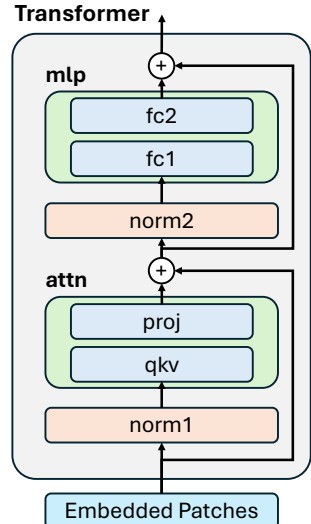

Figure 9: **Transformer Architecture of ViT-B/16.** We describe only the weighted layers, *i.e.*, fully connected layers and layer normalization layers.

Table 10: Pearson correlation coefficient between the accuracy on ImageNet-1K and the dataset distance between ImageNet-1K and each fine-tuning dataset. There is a negative correlation between accuracy and dataset distance. Notably, FT and Prompter consistently exhibit a strong negative correlation across different pretrained models.

| Method | FT | LinearProbing | Visual Prompt | LoRA | EWC | LwF | LP-FT | WiSE-FT | Model Soup |
|---|---|---|---|---|---|---|---|---|---|
| IN-1K + AugReg | -0.64 | -0.22 | -0.91 | -0.63 | -0.57 | -0.49 | -0.59 | -0.46 | -0.54 |
| IN-21K | -0.77 | -0.36 | -0.92 | -0.25 | -0.88 | -0.56 | -0.69 | -0.92 | -0.89 |
| IN-21K + AugReg | -0.68 | 0.10 | -0.86 | -0.63 | -0.91 | -0.38 | -0.39 | -0.52 | -0.51 |
| LAION-2B | -0.67 | -0.19 | -0.74 | -0.31 | -0.32 | -0.44 | -0.56 | -0.31 | -0.13 |

We also measure Normalized Compression Distance (NCD) using both images and the features from ImageNet-1K with AugReg pretrained ViT-B/16. However, the distance between each dataset pair is too insignificant to compare with each dataset as shown in Figures 10c and 10d.

### B.2 Optimal Transport Dataset Distance Aligns With ImageNet-1K Accuracy Drop During Fine-Tuning

We analyze how ImageNet-1K accuracy changes after fine-tuning on downstream datasets. Using the Optimal Transport Dataset Distance (OTDD) (Alvarez-Melis & Fusi, 2020) in the ViT-B/16 feature space, we find that accuracy generally decreases as OTDD from ImageNet-1K increases (Figure 11). Pearson correlations (Table 10) confirm a negative trend for all methods except linear probing, with FT and Visual Prompt showing strong correlations ($< -0.5$). However, OTDD does not consistently correlate with out-of-distribution (OOD) accuracy post-fine-tuning.

## C Fine-Tuning on Small Subset of Fine-Tuning Dataset

Figure 12 shows that CLIP models pretrained on large-scale datasets are particularly vulnerable when fine-tuned on small datasets. To investigate whether this degradation is specific to CLIP or also affects classification models fine-tuned on ImageNet-1K, we compare an ImageNet-1K with AugReg and ImageNet-21K with AugReg pretrained ViT-B/16 models (classification models) with OpenAI and LAION-2B pretrained ViT-B/16 models, both in their original form and after classification fine-tuning. We fine-tune these models on small subsets of each fine-tuning dataset within ImageNet-RIB and evaluate their average accuracy on both

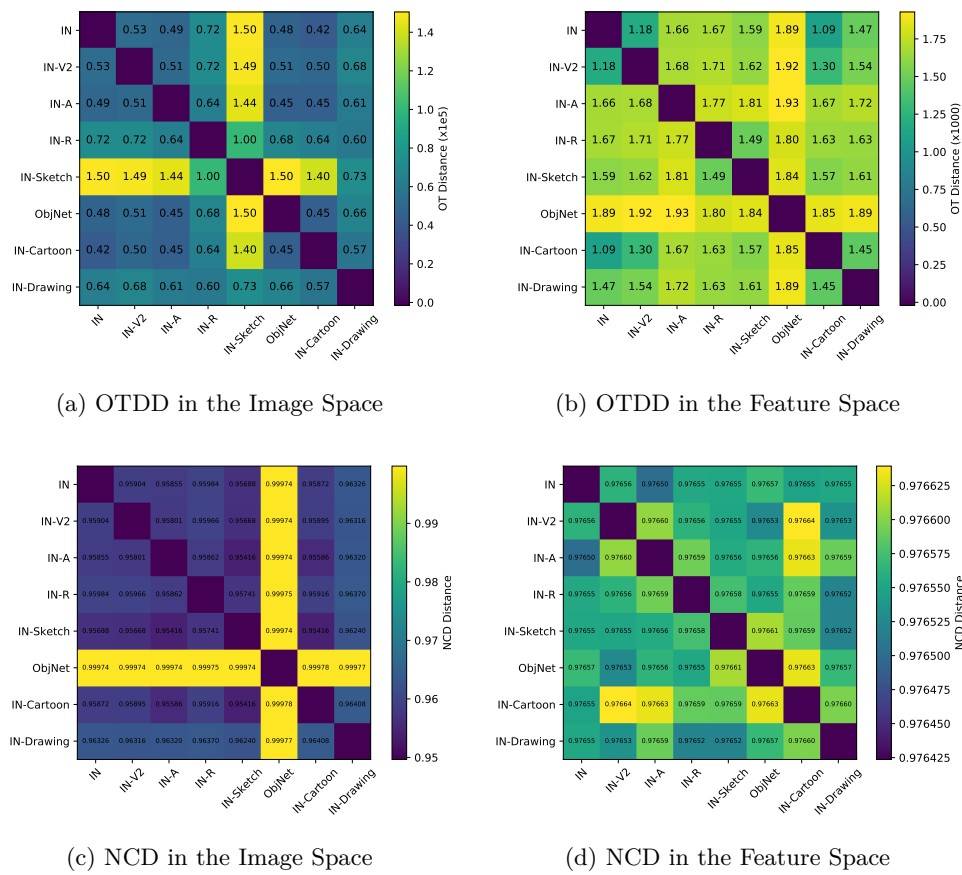

(a) OTDD in the Image Space

(b) OTDD in the Feature Space

(c) NCD in the Image Space

(d) NCD in the Feature Space

Figure 10: **Optimal Transport Dataset Distances (OTDD) in the feature space aligns with each dataset design.** Pairwise OTDD (up) and Normalized Compression Distance (NCD) (down) between datasets using images (left) and features extracted by ImageNet-1K with AugReg pretrained ViT-B/16 on each dataset (right), respectively.

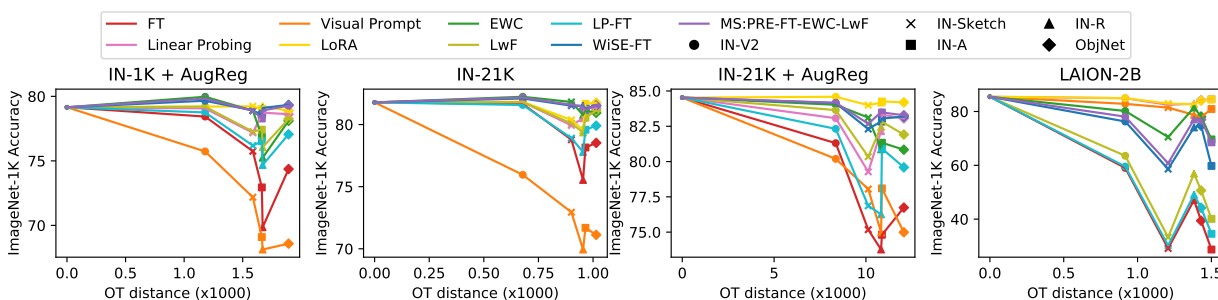

Figure 11: **Relationship between post fine-tuning ImageNet-1K accuracy and the distance between ImageNet-1K and the fine-tuning dataset.** As the distance increases, accuracy generally decreases across fine-tuning methods. We exclude synthetic datasets made from ImageNet-1K validation set to avoid interference.

in-distribution (ID) and out-of-distribution (OOD) datasets. While ImageNet-21K with AugReg pretrained model also experiences a performance drop on the fine-tuning dataset when the number of samples per class falls below 10, the degradation is significantly less severe than that observed for LAION-2B models. On the other hand, ImageNet-1K with AugReg pretrained model's performance increases. As the fine-tuning dataset size increases, the OOD accuracy of classification models gradually declines, indicating a progressive

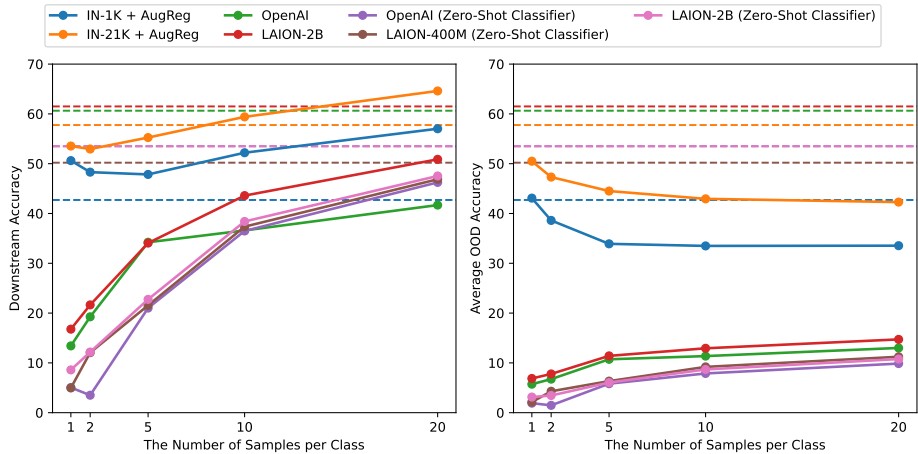

Figure 12: **Fine-tuning on a small dataset leads to severe accuracy degradation in both in- and out-of-distribution.** Average accuracy on fine-tuning datasets and OOD datasets after fine-tuning pretrained ViT-B/16 models on a small number of images per class of fine-tuning datasets. The dashed line denotes the average accuracy of the pretrained models.

Table 11: $mRI$ values obtained using the best validation accuracy for each model on the fine-tuning datasets. Parentheses indicate the accuracy difference compared to models fine-tuned for 10 epochs without splitting the training and validation sets. The average number of epochs needed to achieve the highest validation accuracy on each fine-tuning dataset. †: WiSE-FT and Model Soup are post-hoc weight interpolation methods and do not involve training.

| Method | IN-1K + AugReg | | IN-21K + AugReg | | LAION-2B | |
| | $mRI$ | Best Epoch | $mRI$ | Best Epoch | $mRI$ | Best Epoch |
|---|---|---|---|---|---|---|
| FT | 2.8 (+1.5) | 4.2 | -5.6 (-0.1) | 7.1 | -40.4 (-2.3) | 12.1 |
| Linear Probing | 1.6 (+0.9) | 17.8 | -0.7 (-0.4) | 11.5 | **-2.3 (-0.3)** | 14.1 |
| Visual Prompt | -4.1 (+0.4) | 22.4 | -9.1 (-0.3) | 22.4 | -8.9 (-0.7) | 20.6 |
| LoRA | 2.6 (+1.7) | 20.1 | 1.2 (+3.3) | 16.5 | -2.9 (+0.7) | 15.6 |
| EWC | 3.6 (+0.8) | 18.1 | 0.5 (-0.1) | 19.8 | -14.1 (-1.6) | 24.5 |
| LwF | 4.0 (+0.9) | 3.8 | -1.4 (-0.4) | 4.2 | -34.6 (-0.7) | 11.5 |
| LP-FT | 3.7 (+1.4) | 7.2 | -2.4 (+0.2) | 13.2 | -36.7 (+0.4) | 13.8 |
| WiSE-FT | 4.5 (+0.9) | -† | 1.6 (-0.1) | - | -23.0 (-1.4) | - |
| MS:PRE-FT-EWC-LwF | **4.8 (+0.9)** | - | **2.0 (-0.2)** | - | -19.2 (-1.3) | - |

adaptation to the specific dataset. In contrast, the OOD accuracy of CLIP models and ones fine-tuned on ImageNet-1K initially collapse but then increase with more fine-tuning, suggesting a different adaptation mechanism.

## D ImageNet-RIB with Train-Validation Split

In the original ImageNet-RIB benchmark, the entire fine-tuning dataset is used for fine-tuning. To evaluate the robustness of the models under a different setup, we introduce a train-validation split (4:1 ratio), where models are fine-tuned on the training set, validated on the validation set, and then evaluated on the benchmark using the best-performing epoch from the validation set. We divide the fine-tuning dataset into training and validation sets, extending the training duration to 25 epochs to identify the optimal model based on validation accuracy. Using the best-performing model, we applied robust fine-tuning methods, including LP-FT, WiSE-FT, and Model Soup, to evaluate out-of-distribution (OOD) performance.

For this variant of the benchmark, we showcase results using ViT-B/16 models pretrained on ImageNet-1K with AugReg, ImageNet-21K with AugReg, and LAION-2B. Table 11 reports the mean Robustness

Table 12: mean effective robustness improvement across OOD datasets obtained using the best validation accuracy for each model on the fine-tuning datasets. Parentheses indicate the accuracy difference compared to models fine-tuned for 10 epochs without splitting the training and validation sets. The average number of epochs needed to achieve the highest validation accuracy on each fine-tuning dataset.

| Method | IN-1K + AugReg | IN-21K + AugReg | LAION-2B |
|---|---|---|---|
| FT | -17.6 | -18.5 | -31.7 |
| linear Probing | -11.4 | -9.2 | -9.7 |
| Visual Prompt | -13.6 | -12.8 | -10.6 |
| LoRA | **-3.7** | **-5.4** | **-0.3** |
| EWC | -11.5 | -11.2 | -19.5 |
| LwF | -15.3 | -13.4 | -29.3 |
| LP-FT | -17.3 | -16.4 | -30.9 |
| Wise-FT | -11.6 | -11.6 | -21.9 |
| MS:PRE-FT-EWC-LwF | -12.7 | -12.0 | -21.4 |

Improvement ($mRI$) and the average number of epochs required for each model to achieve its highest validation accuracy on the fine-tuning datasets. Notably, the performance under the train-validation split does not significantly differ from the results in Table 3, where the entire fine-tuning dataset is used for fine-tuning with models trained for 10 epochs.

## E  Effective Robustness

As we mentioned in Section 3, our robust improvement metrics is based on relative robustness. In this section, we employ effective robustness (Taori et al., 2020) instead of direct accuracy difference (relative robustness) to compute effective robustness improvement, $eRI$ (mean effective robustness while fine-tuning on dataset $D_i$):

$$eRI_i = \frac{1}{n-1} \sum_{j=1, j \neq i}^{n} A_i^{(j)} - \beta_j(A_i^{(i)}),$$ (2)

where $\beta_j(x)$ denotes a baseline accuracy on dataset $D_j$ when the accuracy on the dataset $D_i$ is $x$. We calculate mean $eRI$ in ImageNet-RIB with Train-Validation Split (Appendix D). Note that in the original ImageNet-RIB setting, we use the entire dataset for training, not splitting the validation set. Table 12 demonstrates that all mean $eRI$ become negative. This is because, unlike previous benchmark (Shi et al., 2023; Taori et al., 2020) where the downstream dataset (ImageNet-1K) contains all labels in OOD datasets, our downstream dataset does not contain all labels (*e.g.*, ImageNet-R has 200 classes while ImageNet-Sketch has 1000 classes). That is why the robustness can be negative. Even under this condition, the LAION-2B retrained model with FT performs much worse than others.

## F  Experimental Details

In this section, we describe the details of the experimental setup. We use a single NVIDIA RTX 4090 GPU for the experiment.

### F.1  Out-of-Distribution Datasets in ImageNet-RIB

We leverage all existing ImageNet variants designed to measure the robustness of the trained network during distribution shifts. ImageNet-O (Hendrycks et al., 2021b) is not used since it is an out-of-distribution detection dataset.

Table 13: Python libraries and the names of network weights for each pretrained model.

| Architecture | Pretraining Dataset | Library | Weight Name |
|---|---|---|---|
| ViT-B/16 | IN-1K + AugReg | timm | vit_base_patch16_224.augreg_in1k |
| | IN-1K + SAM | timm | vit_base_patch16_224.sam_in1k |
| | IN-21K | timm | vit_base_patch16_224.orig_in21k_ft_in1k |
| | IN-21K + AugReg | timm | vit_base_patch16_224.augreg_in21k_ft_in1k |
| | IN-21K-P | timm | vit_base_patch16_224_miil.in21k_ft_in1k |
| | LAION-2B | timm | vit_base_patch16_clip_224.laion2b_ft_in1k |
| | OpenAI | timm | vit_base_patch16_clip_224.openai_ft_in1k |
| ViT-B/32 | IN-1K + AugReg | timm | vit_base_patch32_224.augreg_in1k |
| | IN-21K + AugReg | timm | vit_base_patch32_224.augreg_in21k_ft_in1k |
| | LAION-2B | timm | vit_base_patch32_clip_224.laion2b_ft_in1k |
| | OpenAI | timm | vit_base_patch32_clip_224.openai_ft_in1k |
| ViT-S/16 | IN-1K + AugReg | timm | vit_small_patch16_224.augreg_in1k |
| | IN-21K + AugReg | timm | vit_small_patch16_224.augreg_in21k_ft_in1k |
| ViT-S/32 | IN-21K + AugReg | timm | vit_small_patch32_224.augreg_in21k_ft_in1k |
| ViT-L/16 | IN-21K + AugReg | timm | vit_large_patch16_224.augreg_in21k_ft_in1k |
| ResNet-18 | IN-1K | torchvision | ResNet18_Weights.DEFAULT |
| ResNet-50 | IN-1K | torchvision | ResNet50_Weights.DEFAULT |

**ImageNet-V2 (Recht et al., 2019)** ImageNet-V2 is designed to have a distribution as similar as possible to the original ImageNet-1K. It has 50,000 images with 1,000 classes same as the original validation set. The dataset is used under the MIT license.

**ImageNet-A (Hendrycks et al., 2021b)** ImageNet-A is an adversarially filtered test image that ImageNet-1K pretrained ResNet-50 (He et al., 2016) is difficult to predict correctly. It contains 7,500 images with 200 difficult subclasses from ImageNet-1K. The dataset is used under the MIT license.

**ImageNet-R (Hendrycks et al., 2021a)** ImageNet-R (Renditions) contains 30,000 images from 200 ImageNet classes with various rendition styles such as painting, sculpture, embroidery, origami, cartoon, toy, and so on. The drawing rendition overlaps with ImageNet-Sketch (Wang et al., 2019). The dataset is used under the MIT license.

**ImageNet-Sketch (Wang et al., 2019)** ImageNet-Sketch comprises black and white sketch drawings of the ImageNet-1K classes and each class has 50 images. The dataset is used under the MIT license.

**ImageNet-Cartoon and ImageNet-Drawing (Salvador & Oberman, 2022)** ImageNet-Cartoon and ImageNet-Drawing are to be converted from ImageNet validation set images to cartoon, and drawing styles based on generative adversarial network (Wang & Yu, 2020) and image processing (Lu et al., 2012). These simplified representations test a model's ability to identify objects from minimalistic and abstract visual information. The dataset is used under the Creative Commons Attribution 4.0 International license.

**ObjectNet (Barbu et al., 2019)** ObjectNet is designed for evaluating object recognition models under more realistic conditions such as various poses, backgrounds, and viewpoints. There are 50,000 images with 313 object classes and 113 classes are overlapped with ImageNet. We only use ImageNet class objects. The dataset is used under the MIT license.

**ImageNet-C (Hendrycks & Dietterich, 2019)** ImageNet-C is designed for measuring the robustness of models to common perturbations such as noise, blur, weather, and digital distortions. In the dataset, ImageNet validation set images are perturbed with various severity from 1 to 5. Unlike the original metrics, corruption error compared with AlexNet, we use average accuracy for consistency with other datasets. The dataset is used under the Apache-2.0 license.

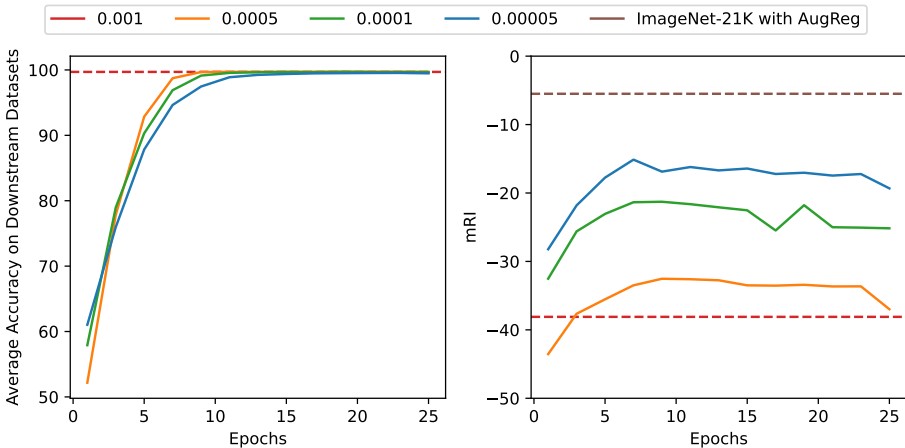

Figure 13: $mRI$ **is much lower than ImageNet-21K with AugReg model, even with a small learning rate.** Average accuracy on the fine-tuning datasets and $mRI$ while fine-tuning pretrained ViT-B/16 models.

## F.2 Pretrained Model

Table 13 lists the libraries and corresponding network weight names for each model. We use the entire models in timm and torchvision library, which are finally fine-tuned on ImageNet-1K, with patch sizes of 16 and 32, and input image shape of 224 among ViT small, base, and large. For ResNets, we use the default ImageNet-1K pretrained weights from the torchvision library.

## F.3 Training and Hyperparameters

Each pretrained model is fine-tuned on the downstream dataset for 10 epochs where the average accuracy on fine-tuning datasets for each pretrained ViT-B/16 model achieves more than 90% with vanilla fine-tuning. We applied LoRA on query and value projection layers with rank 8 following the original implementation (Hu et al., 2021). We use 2 as a temperature for calculating KL divergence for LwF following Li & Hoiem (2017). For WiSE-FT, we use the interpolation ratio between pretrained and fine-tuned models as 0.5 following the recommendation by Wortsman et al. (2022b) instead of finding the best hyperparameters evaluated on the benchmark for the fair comparison. Appendix G.4 compares with results of the best-performing ratio.

# G Ablation Studies

## G.1 Catastrophic Forgetting Persists Despite Small Learning Rates

To test whether catastrophic forgetting stems solely from large learning rates, we fine-tune the LAION-2B pretrained ViT-B/16 on fine-tuning datasets using reduced learning rates (0.0005, 0.0001, 0.00005). We extend training to 25 epochs to account for slower learning rates and evaluate performance throughout training (Figure 13). Although a smallr learning rate attenuates forgetting, the native $mRI$ remains substantially higher than that of the ImageNet-21K AugReg baseline (-5.5; Table 3). This implies that smaller learning rate is not a solution for the severe robustness degradation.

## G.2 Weight Decays

Beyond the learning rate analysis mentioned in Appendix G.1, we also fine-tune LAION-2B pre-trained VIT-B/16 with various weight decays in Table 14. Across all settings, $mRI$ is substantially lower than ViT-B/16 pretrained on smaller datasets in Table 3.

Table 14: $mRI$ of ViT-B/16 pre-trained on LAION-2B with various weight decay used in fine-tuning. We use vanilla fine-tuning (FT).

| Weight Decay | 0 | 0.0001 | 0.0005 | 0.001 | 0.005 | 0.01 | 0.1 |
|---|---|---|---|---|---|---|---|
| $mRI$ | **-38.1** | -39.1 | -44.2 | -39.4 | -40.9 | -49.4 | -59.3 |

Table 15: $mRI$ of ViT-B/16 pre-trained on three different datasets after fine-tuning with different fine-tuning methods. We run with three random seeds and calculate the mean and standard error.

| Method | ImageNet 1K + AugReg | ImageNet 21K + AugReg | LAION-2B |
|---|---|---|---|
| FT | 1.4 ± 0.0 | -5.4 ± 0.1 | -38.2 ± 0.0 |
| Linear Probing | 0.8 ± 0.0 | -0.3 ± 0.0 | -2.0 ± 0.1 |
| EWC | 2.9 ± 0.1 | 0.8 ± 0.1 | -12.9 ± 0.4 |
| LwF | 3.2 ± 0.1 | -0.8 ± 0.1 | -33.5 ± 0.2 |
| LP-FT | 2.0 ± 0.2 | -2.7 ± 0.2 | -37.2 ± 0.1 |
| Wise-FT | 3.7 ± 0.1 | 1.9 ± 0.1 | -21.5 ± 0.3 |
| MS: PRE-FT-EWC-LwF | 4.0 ± 0.1 | 2.3 ± 0.1 | -18.0 ± 0.2 |

### G.3 Robustness to Multiple Random Seeds

We test whether the results vary depending on the random seeds. Table 15 illustrates $mRI$ of three different pretrained ViT-B/16. As the data clearly shows, the standard errors across all runs are very small. This confirms that our findings are highly stable and not an artifact of a particular random seed. Most importantly, the central conclusions of our paper are fully supported by this statistical analysis. For the vanilla fine-tuning method, there remains a massive gap in post-tuning robustness (mRI) between the LAION-2B model and the models pretrained on ImageNet.

### G.4 The Best Ratio for WiSE-FT

We conduct a grid search from 0.1 to 0.9 with an increment of 0.1 to find the best-performing ratio ($\alpha$) between the pretrained ViT-B/16's weight and the fine-tuned model's weight for WiSE-FT:

$$\mathbf{W}_{\text{WiSE-FT}} = \alpha \cdot \mathbf{W}_{\text{pre}} + (1 - \alpha) \cdot \mathbf{W}_{\text{FT}}, \tag{3}$$

where $\mathbf{W}_{\text{WiSE-FT}}$, $\mathbf{W}_{\text{pre}}$, and $\mathbf{W}_{\text{FT}}$ represent the network weights of WiSE-FT, the pretrained model, and the vanilla fine-tuned model, respectively.

It is important to note that this hyperparameter search, based on test results ($mRI$), constitutes an unfair comparison with other methods. Table 16 compares the $mRI$ achieved by WiSE-FT using the default ratio of 0.5 from WiSE-FT (Wortsman et al., 2022b) and the best ratio. WiSE-FT using OpenAI or LAION-2B pretrained models performs significantly better with the best ratio, as it relies minimally on the fine-tuned model's weights. Similarly, hyperparameter search for Model Soup, which combines network weights from pretrained and fine-tuned models (e.g., FT, EWC, LwF), could further improve performance. Notably, WiSE-FT is a special case of Model Soup when the ratios for EWC and LwF are set to 0. However, exploring the optimal ratio for Model Soup weights is beyond the scope of this study.

## H Additional Experiments with Various pretrained Models

### H.1 Robustness of pretrained Models

We evaluate pretrained models mentioned in Appendix F.2 on OOD datasets as shown in Table 17. Larger networks with smaller patch sizes achieve higher accuracy on both ImageNet-1K and OOD datasets. Similarly, models pretrained on larger, more diverse datasets demonstrate better performance.

Table 16: $mRI$ of ViT-B/16 pretrained on various datasets using WiSE-FT with default ratio (0.5) and the best ratio

| Pretraining Dataset | WiSE-FT ($\alpha = 0.5$) | WiSE-FT (best $\alpha$) | best $\alpha$ |
|---|---|---|---|
| IN-1K + AugReg | 3.6 | 4.7 | 0.4 |
| IN-1K + SAM | 3.6 | 4.1 | 0.3 |
| IN-21K | 2.5 | 2.5 | 0.5 |
| IN-21K-P | 3.0 | 3.0 | 0.5 |
| IN-21K + AugReg | 1.7 | 2.3 | 0.7 |
| OpenAI | -18.1 | -1.6 | 0.9 |
| LAION-2B | -21.6 | -2.4 | 0.9 |

Table 17: The average accuracy of various pretrained models on ImageNet-1K validation set and OOD datasets.

| Arch. | Pretraining Dataset | ImageNet-1K | IN-V2 | IN-A | IN-R | IN-Sketch | ObjNet | IN-Cartoon | IN-Drawing | IN-C |
|---|---|---|---|---|---|---|---|---|---|---|
| | IN-1K + AugReg | 79.2 | 66.4 | 15.0 | 38.0 | 28.0 | 33.0 | 66.2 | 39.1 | 56.0 |
| | IN-1K + SAM | 80.2 | 68.2 | 9.0 | 40.1 | 27.7 | 34.2 | 66.9 | 42.3 | 54.6 |
| | IN-21K | 81.8 | 71.4 | 32.0 | 47.3 | 35.8 | 42.5 | 69.4 | 44.1 | 58.3 |
| ViT-B/16 | IN-21K-P | 84.3 | 74.0 | 34.1 | 51.5 | 40.2 | 46.7 | 73.5 | 45.1 | 61.4 |
| | IN-21K + AugReg | 84.5 | 74.0 | 43.2 | 56.8 | 43.2 | 48.4 | 75.1 | 54.9 | 66.5 |
| | OpenAI | 85.3 | 75.7 | 47.3 | 65.9 | 50.9 | 50.7 | 76.3 | 55.7 | 62.6 |
| | LAION-2B | 85.5 | 75.6 | 41.5 | **68.8** | **55.4** | 51.1 | 78.2 | 58.4 | 63.0 |
| | IN-1K + SAM | 73.7 | 59.9 | 4.3 | 36.6 | 23.0 | 25.2 | 63.2 | 40.6 | 48.8 |
| ViT-B/32 | IN-21K + AugReg | 80.7 | 69.0 | 22.4 | 49.3 | 37.1 | 40.7 | 70.6 | 42.5 | 60.5 |
| | OpenAI | 82.0 | 70.9 | 22.6 | 55.8 | 45.0 | 41.5 | 71.1 | 42.5 | 57.9 |
| | LAION-2B | 82.6 | 71.6 | 22.8 | 59.2 | 49.1 | 43.5 | 73.0 | 42.3 | 57.5 |
| ViT-S/16 | IN-1K + AugReg | 78.8 | 66.7 | 13.4 | 37.1 | 25.9 | 25.2 | 63.3 | 37.2 | 53.2 |
| | IN-21K + AugReg | 81.4 | 70.3 | 27.0 | 46.0 | 32.9 | 32.2 | 67.8 | 37.7 | 58.0 |
| ViT-S/32 | IN-21K + AugReg | 76.0 | 63.9 | 11.5 | 39.7 | 26.2 | 24.8 | 62.9 | 34.3 | 52.0 |
| ViT-L/16 | IN-21K + AugReg | **85.8** | **76.2** | **55.5** | 64.4 | 51.8 | **52.8** | **79.5** | **64.6** | **72.2** |
| ResNet-18 | IN-1K | 69.8 | 57.3 | 1.1 | 33.1 | 20.2 | 18.1 | 48.2 | 20.4 | 31.7 |
| ResNet-50 | IN-1K | 80.3 | 69.5 | 16.7 | 41.6 | 28.4 | 33.0 | 61.1 | 31.1 | 46.6 |

Table 18: Average backward transfer on the ImageNet-1K validation set for each method, evaluated across different architectures and pretraining datasets. Bold indicates the highest backward transfer for each model.

| Architecture | ViT-B/16 | | | | | | | ViT-B/32 | | | | | ViT-S/16 | | ViT-S/32 | ViT-L/16 | ResNet-18 | ResNet-50 |
|---|---|---|---|---|---|---|---|---|---|---|---|---|---|---|---|---|---|---|
| Method | IN-1K + AugReg | IN-1K + SAM | IN-21K | IN-21K-P | IN-21K + AugReg | OpenAI | LAION-2B | IN-1K + AugReg | IN-1K + SAM | IN-21K + AugReg | OpenAI | LAION-2B | IN-1K + AugReg | IN-21K + AugReg | IN-21K + AugReg | IN-21K + AugReg | IN-1K | IN-1K |
| FT | 0.6 | 1.3 | 0.6 | 0.8 | -1.0 | -28.0 | -28.5 | 0.5 | 1.7 | 0.4 | -27.5 | -31.7 | -1.3 | -3.0 | -2.2 | 1.5 | -7.9 | -7.3 |
| Linear Probing | 1.8 | 1.5 | 1.5 | 1.6 | 1.6 | **0.0** | **0.3** | 2.7 | 2.4 | 2.2 | **-0.4** | **-0.3** | 1.4 | 1.3 | 1.6 | 1.9 | -13.6 | -19.4 |
| Visual Prompt | -6.9 | -7.6 | -8.1 | -7.3 | -6.0 | -4.5 | -4.3 | -9.2 | -9.0 | -9.3 | -6.8 | -6.7 | -9.3 | -9.0 | -13.7 | -8.2 | -16.1 | -6.4 |
| LoRA | -0.1 | -0.1 | -0.6 | -0.1 | -2.7 | -2.7 | -2.5 | -0.4 | -0.1 | -0.2 | -2.8 | -2.5 | -1.5 | -0.3 | -0.3 | 0.1 | - | - |
| EWC | -0.7 | 0.0 | -0.3 | -0.6 | -1.5 | -7.5 | -6.7 | -1.3 | -0.1 | -0.9 | -7.1 | -9.2 | -1.2 | -2.0 | -2.0 | -0.6 | -12.4 | -16.6 |
| LwF | **3.2** | **2.1** | **2.3** | **2.3** | **3.2** | -21.0 | -21.8 | **3.2** | **2.9** | **3.1** | -18.9 | -22.0 | **2.8** | **2.6** | **2.9** | **3.6** | -1.3 | -10.3 |
| LP-FT | 2.0 | 0.8 | 1.3 | 1.8 | 1.1 | -25.6 | -25.9 | 2.9 | 2.2 | 2.5 | -25.1 | -29.2 | 0.9 | 0.1 | 0.9 | -2.1 | -6.8 | -6.6 |
| WiSE-FT | 2.7 | 1.7 | 1.7 | 1.8 | 2.2 | -7.1 | -9.6 | 3.0 | 2.4 | 2.6 | -5.5 | -8.3 | 2.1 | 2.5 | 2.6 | 2.4 | **0.9** | **0.9** |
| MS | 2.6 | 1.7 | 1.9 | 1.7 | 2.4 | -6.7 | -7.3 | 2.7 | 2.4 | 2.3 | -4.8 | -6.9 | 2.1 | 2.2 | 2.3 | 2.6 | -0.8 | -2.0 |

## H.2  Backward Transfer

We measure the backward transfer, accuracy change on the pretraining dataset, ImageNet-1K validation set after fine-tuning. Table 18 presents the average backward transfer across different fine-tuning methods on downstream datasets. While LwF achieves the best backward transfer in most models, linear probing outperforms it on OpenAI and LAION-2B pretrained models.

## H.3   Performance on Fine-Tuning Dataset

Tables 19  20, and  21 demonstrate the accuracy on fine-tuning datasets (*i.e.*, training accuracy) with ViT base, ViT large and ViT small, and ResNet, respectively. FT, LwF, and LP-FT can overfit to the fine-tuning dataset but WiSE-FT and Model Soup (PRE-FT-LwF-EWC) have worse performance which might be due to using pretrained model weights. Visual Prompt and LoRA rarely learn from a fine-tuning dataset.

Table 19: Training accuracy on downstream datasets after fine-tuning with each method using ViT-B/16. FT and LP-FT generally achieve the highest performance, while Visual Prompt and LoRA show the lowest.

| Arch. | Pretraining Dataset | Method | IN-V2 | IN-A | IN-R | IN-Sketch | ObjNet | IN-Cartoon | IN-Drawing | IN-C |
|---|---|---|---|---|---|---|---|---|---|---|
| ViT-B/16 | IN-1K + AugReg | FT | 96.3 | **97.5** | 98.4 | 96.0 | 97.7 | 97.5 | 97.6 | **100.0** |
| | | Linear Probing | 71.5 | 42.4 | 60.4 | 58.8 | 57.8 | 76.1 | 61.2 | 89.7 |
| | | Visual Prompt | 66.6 | 28.8 | 58.9 | 45.8 | 46.3 | 72.7 | 64.8 | 64.4 |
| | | LoRA | 66.5 | 18.7 | 41.8 | 39.0 | 36.7 | 69.7 | 62.4 | 58.6 |
| | | EWC | 72.0 | 54.3 | 65.3 | 50.3 | 50.8 | 76.2 | 70.3 | 67.7 |
| | | LwF | 95.8 | 95.5 | 97.3 | 95.1 | 95.1 | 96.7 | 96.5 | 100.0 |
| | | LP-FT | **96.7** | 97.2 | **98.5** | **96.1** | **97.9** | **97.5** | **97.6** | 94.6 |
| | | WiSE-FT | 81.2 | 56.6 | 71.2 | 61.4 | 63.7 | 84.0 | 74.0 | 88.8 |
| | | MS:PRE-FT-EWC-LwF | 82.4 | 67.4 | 75.5 | 66.8 | 66.8 | 85.5 | 78.7 | 88.0 |
| | IN-1K + SAM | FT | 77.9 | 67.2 | **87.2** | 84.3 | **75.1** | 87.1 | 85.7 | **100.0** |
| | | Linear Probing | 68.7 | 14.3 | 50.5 | 38.8 | 41.4 | 71.3 | 53.6 | 80.7 |
| | | Visual Prompt | 64.4 | 17.0 | 50.6 | 37.2 | 40.1 | 69.7 | 56.2 | 57.7 |
| | | LoRA | 68.2 | 10.0 | 44.9 | 32.7 | 36.7 | 69.6 | 49.6 | 67.5 |
| | | EWC | 69.0 | 23.9 | 50.4 | 43.8 | 41.3 | 72.6 | 62.3 | 59.6 |
| | | LwF | 77.6 | 62.5 | 84.2 | 81.7 | 69.7 | 85.9 | 84.0 | 99.9 |
| | | LP-FT | **78.3** | 64.9 | 86.6 | 83.5 | 74.6 | **87.2** | **86.1** | 84.4 |
| | | WiSE-FT | 72.7 | 31.4 | 64.7 | 52.6 | 52.4 | 78.9 | 68.1 | 78.8 |
| | | MS:PRE-FT-EWC-LwF | 72.8 | 36.5 | 66.7 | 55.6 | 53.0 | 79.3 | 70.7 | 80.3 |
| | IN-21K | FT | 92.2 | 94.9 | **96.3** | 92.8 | 94.3 | 94.7 | 94.1 | 100.0 |
| | | Linear Probing | 75.0 | 51.8 | 66.4 | 59.0 | 63.2 | 77.7 | 59.6 | 86.4 |
| | | Visual Prompt | 66.8 | 37.4 | 58.2 | 43.9 | 51.0 | 68.9 | 57.9 | 58.6 |
| | | LoRA | 71.5 | 38.2 | 52.9 | 39.8 | 47.1 | 73.5 | 53.8 | 49.4 |
| | | EWC | 74.5 | 59.7 | 65.6 | 50.1 | 56.1 | 77.3 | 67.6 | 66.5 |
| | | LwF | 91.9 | 92.8 | 94.3 | 90.9 | 91.2 | 93.7 | 92.1 | 99.9 |
| | | LP-FT | **93.4** | **95.1** | 96.2 | **93.1** | **94.7** | **95.1** | **94.3** | 97.3 |
| | | WiSE-FT | 81.8 | 67.7 | 75.1 | 63.5 | 68.7 | 83.2 | 72.8 | 84.7 |
| | | MS:PRE-FT-EWC-LwF | 82.6 | 73.7 | 78.3 | 67.0 | 70.6 | 84.5 | 76.0 | 88.7 |
| | IN-21K-P | FT | 95.4 | 98.7 | 99.3 | 96.7 | 99.2 | 97.3 | 97.6 | **100.0** |
| | | Linear Probing | 78.0 | 57.0 | 70.5 | 67.3 | 68.5 | 81.0 | 64.5 | 88.8 |
| | | Visual Prompt | 70.2 | 43.1 | 63.3 | 49.9 | 56.1 | 74.6 | 63.6 | 63.2 |
| | | LoRA | 74.2 | 37.5 | 53.1 | 47.4 | 48.9 | 75.6 | 67.4 | 63.1 |
| | | EWC | 76.8 | 66.7 | 73.0 | 57.8 | 61.0 | 80.7 | 73.9 | 69.7 |
| | | LwF | 94.2 | 97.2 | 98.5 | 95.7 | 97.0 | 96.1 | 96.2 | 100.0 |
| | | LP-FT | **96.2** | **98.8** | **99.4** | **96.9** | **99.3** | **97.7** | **98.1** | 100.0 |
| | | WiSE-FT | 84.2 | 74.3 | 80.1 | 70.2 | 73.4 | 87.0 | 78.0 | 88.7 |
| | | MS:PRE-FT-EWC-LwF | 84.7 | 80.8 | 82.8 | 73.5 | 75.9 | 87.8 | 80.9 | 89.0 |
| | IN-21K + AugReg | FT | 100.0 | **100.0** | **99.8** | 98.0 | **100.0** | 99.9 | 99.9 | **100.0** |
| | | Linear Probing | 98.3 | 97.3 | 91.7 | 93.2 | 92.0 | 96.5 | 91.1 | 98.6 |
| | | Visual Prompt | 74.2 | 52.0 | 71.7 | 56.6 | 61.1 | 78.7 | 73.2 | 70.2 |
| | | LoRA | 75.1 | 53.1 | 66.5 | 56.5 | 56.4 | 78.6 | 74.4 | 19.2 |
| | | EWC | 91.1 | 97.8 | 91.2 | 73.4 | 93.8 | 86.2 | 84.1 | 76.8 |
| | | LwF | 100.0 | **100.0** | 99.8 | 98.0 | **100.0** | 99.9 | 99.9 | 100.0 |
| | | LP-FT | **100.0** | **100.0** | **99.8** | **98.1** | **100.0** | 99.9 | **99.9** | 100.0 |
| | | WiSE-FT | 95.9 | 97.0 | 94.7 | 88.1 | 91.0 | 95.3 | 92.7 | 96.2 |
| | | MS:PRE-FT-EWC-LwF | 96.8 | 98.6 | 96.5 | 89.9 | 95.9 | 95.3 | 93.9 | 96.8 |
| | OpenAI | FT | **100.0** | **100.0** | 99.8 | **98.0** | **100.0** | **99.9** | 99.9 | **100.0** |
| | | Linear Probing | 82.3 | 78.0 | 86.1 | 74.1 | 79.3 | 86.0 | 79.5 | 92.2 |
| | | Visual Prompt | 77.7 | 54.4 | 76.9 | 58.1 | 60.4 | 80.3 | 71.2 | 66.5 |
| | | LoRA | 79.1 | 65.1 | 79.2 | 60.1 | 62.0 | 83.0 | 76.9 | 41.7 |
| | | EWC | 88.7 | 90.0 | 90.9 | 73.8 | 86.2 | 87.0 | 85.4 | 77.8 |
| | | LwF | **100.0** | 100.0 | 99.8 | 98.0 | 99.9 | 99.9 | 99.9 | **100.0** |
| | | LP-FT | 100.0 | **100.0** | **99.8** | 98.0 | **100.0** | 99.9 | **99.9** | 100.0 |
| | | WiSE-FT | 88.0 | 76.8 | 89.9 | 78.6 | 81.5 | 91.5 | 91.0 | 94.7 |
| | | MS:PRE-FT-EWC-LwF | 88.9 | 81.7 | 91.3 | 79.4 | 83.3 | 91.0 | 91.1 | 93.0 |
| | LAION-2B | FT | 100.0 | 100.0 | **99.8** | 98.0 | 100.0 | 99.9 | **99.9** | **100.0** |
| | | Linear Probing | 82.8 | 77.2 | 88.4 | 79.3 | 80.9 | 87.6 | 80.0 | 93.3 |
| | | Visual Prompt | 77.2 | 49.9 | 79.6 | 62.1 | 63.6 | 81.3 | 72.4 | 68.1 |
| | | LoRA | 78.1 | 58.6 | 79.8 | 62.3 | 61.5 | 83.9 | 76.4 | 39.8 |
| | | EWC | 83.8 | 68.7 | 89.3 | 71.8 | 79.9 | 86.3 | 83.5 | 74.2 |
| | | LwF | 100.0 | 99.9 | 99.8 | 98.0 | 99.9 | 99.9 | 99.9 | 100.0 |
| | | LP-FT | **100.0** | **100.0** | 99.8 | **98.0** | **100.0** | 99.9 | **99.9** | 100.0 |
| | | WiSE-FT | 85.8 | 46.5 | 87.6 | 77.9 | 77.6 | 91.0 | 89.9 | 93.3 |
| | | MS:PRE-FT-EWC-LwF | 87.3 | 64.6 | 89.5 | 79.2 | 80.3 | 90.6 | 90.1 | 94.3 |

Table 20: Training accuracy on downstream datasets after fine-tuning with each method using various ViTs. FT and LP-FT generally achieve the highest performance, while Visual Prompt and LoRA show the lowest.

| Arch. | Pretraining Dataset | Method | IN-V2 | IN-A | IN-R | IN-Sketch | ObjNet | IN-Cartoon | IN-Drawing | IN-C |
|---|---|---|---|---|---|---|---|---|---|---|
| ViT-B/32 | IN-1K + AugReg | FT | 94.6 | 94.0 | 97.3 | 95.3 | 95.9 | 96.3 | 96.3 | 100.0 |
| | | Linear Probing | 68.7 | 34.9 | 63.4 | 62.1 | 54.7 | 75.1 | 62.6 | 90.9 |
| | | Visual Prompt | 59.6 | 15.1 | 54.0 | 41.4 | 38.3 | 68.3 | 60.1 | 59.3 |
| | | LoRA | 61.1 | 10.1 | 42.1 | 31.9 | 31.0 | 67.3 | 53.1 | 66.3 |
| | | EWC | 65.9 | 33.4 | 59.3 | 45.6 | 42.1 | 71.3 | 64.3 | 62.7 |
| | | LwF | 94.0 | 91.2 | 95.7 | 94.2 | 92.6 | 95.5 | 95.0 | 99.9 |
| | | LP-FT | **96.1** | **94.5** | **97.7** | **95.7** | **96.7** | **96.9** | **96.9** | **100.0** |
| | | WiSE-FT | 77.7 | 39.3 | 67.4 | 58.7 | 56.4 | 81.6 | 71.7 | 87.7 |
| | | MS:PRE-FT-EWC-LwF | 79.0 | 49.1 | 71.0 | 63.0 | 60.6 | 82.7 | 75.4 | 86.8 |
| | IN-1K + SAM | FT | 73.5 | 46.8 | **82.5** | **83.8** | **66.4** | 84.3 | 82.5 | **100.0** |
| | | Linear Probing | 60.7 | 8.9 | 48.3 | 35.5 | 33.4 | 67.2 | 51.2 | 83.4 |
| | | Visual Prompt | 57.2 | 8.5 | 44.6 | 31.8 | 29.9 | 63.9 | 50.1 | 52.8 |
| | | LoRA | 59.9 | 5.2 | 41.6 | 28.1 | 28.6 | 65.3 | 46.9 | 62.2 |
| | | EWC | 60.9 | 10.2 | 45.1 | 37.7 | 31.1 | 67.0 | 52.3 | 53.7 |
| | | LwF | 73.2 | 43.1 | 79.2 | 81.3 | 61.2 | 83.0 | 80.4 | 99.9 |
| | | LP-FT | **74.2** | 44.8 | 81.8 | 82.4 | 66.1 | **84.8** | **82.7** | 100.0 |
| | | WiSE-FT | 66.0 | 17.6 | 59.3 | 46.9 | 42.6 | 74.5 | 64.0 | 76.0 |
| | | MS:PRE-FT-EWC-LwF | 66.1 | 20.1 | 60.8 | 51.1 | 43.3 | 74.8 | 65.5 | 76.5 |
| | IN-21K + AugReg | FT | 99.5 | 100.0 | 99.8 | 97.7 | 99.9 | 99.5 | 99.6 | 100.0 |
| | | Linear Probing | 83.5 | 65.6 | 77.5 | 78.2 | 73.5 | 86.0 | 72.9 | 94.4 |
| | | Visual Prompt | 68.2 | 32.0 | 66.3 | 51.4 | 52.4 | 74.0 | 67.3 | 65.2 |
| | | LoRA | 69.1 | 25.0 | 51.5 | 43.0 | 43.4 | 71.9 | 63.2 | 66.1 |
| | | EWC | 76.3 | 69.5 | 72.1 | 56.3 | 70.0 | 78.0 | 72.5 | 68.0 |
| | | LwF | 99.2 | 99.8 | 99.5 | 97.3 | 99.7 | 99.2 | 99.1 | 100.0 |
| | | LP-FT | **99.8** | **100.0** | **99.8** | **97.9** | **100.0** | **99.8** | **99.8** | **100.0** |
| | | WiSE-FT | 87.9 | 72.0 | 80.4 | 72.3 | 73.8 | 88.9 | 79.1 | 92.5 |
| | | MS:PRE-FT-EWC-LwF | 89.0 | 82.8 | 85.0 | 76.8 | 82.1 | 89.6 | 83.2 | 90.6 |
| | OpenAI | FT | 100.0 | 100.0 | 99.8 | 98.0 | 100.0 | 99.9 | **99.9** | 100.0 |
| | | Linear Probing | 74.8 | 47.1 | 75.9 | 64.3 | 63.6 | 80.1 | 71.2 | 89.0 |
| | | Visual Prompt | 71.6 | 29.3 | 65.6 | 50.7 | 47.5 | 75.5 | 64.9 | 62.3 |
| | | LoRA | 72.5 | 34.7 | 67.7 | 53.3 | 48.4 | 77.9 | 69.6 | 71.5 |
| | | EWC | 88.4 | 86.9 | 88.4 | 70.8 | 79.8 | 85.1 | 83.4 | 72.6 |
| | | LwF | 99.9 | 99.8 | 99.8 | 97.9 | 99.8 | 99.8 | 99.9 | 100.0 |
| | | LP-FT | **100.0** | **100.0** | **99.8** | **98.0** | **100.0** | **99.9** | 99.9 | 100.0 |
| | | WiSE-FT | 85.9 | 71.4 | 88.6 | 78.1 | 75.5 | 89.3 | 89.9 | 93.4 |
| | | MS:PRE-FT-EWC-LwF | 87.0 | 76.0 | 89.1 | 77.7 | 76.6 | 88.1 | 89.5 | 91.1 |
| | LAION-2B | FT | **100.0** | **100.0** | 99.8 | 98.0 | **100.0** | 99.9 | **99.9** | **100.0** |
| | | Linear Probing | 75.5 | 47.8 | 78.7 | 67.3 | 66.7 | 81.4 | 71.5 | 88.7 |
| | | Visual Prompt | 72.2 | 30.0 | 69.2 | 54.6 | 51.2 | 76.1 | 65.1 | 62.1 |
| | | LoRA | 72.9 | 35.7 | 70.4 | 56.4 | 51.4 | 79.3 | 69.5 | 73.4 |
| | | EWC | 85.6 | 80.7 | 87.3 | 71.6 | 77.2 | 84.8 | 83.4 | 69.7 |
| | | LwF | 99.9 | 99.8 | 99.8 | 97.9 | 99.8 | 99.8 | 99.8 | 100.0 |
| | | LP-FT | **100.0** | **100.0** | 99.8 | 98.0 | 100.0 | **99.9** | 99.9 | 100.0 |
| | | WiSE-FT | 85.3 | 68.0 | 87.9 | 77.8 | 76.1 | 87.5 | 88.6 | 92.9 |
| | | MS:PRE-FT-EWC-LwF | 85.9 | 73.9 | 88.8 | 77.8 | 77.1 | 86.4 | 88.5 | 91.9 |
| ViT-S/16 | IN-1K + AugReg | FT | **99.8** | 100.0 | 99.8 | 97.8 | 100.0 | 99.7 | 99.7 | **100.0** |
| | | Linear Probing | 75.3 | 46.2 | 63.2 | 64.0 | 62.4 | 77.7 | 63.8 | 83.5 |
| | | Visual Prompt | 66.2 | 30.7 | 58.6 | 43.6 | 49.2 | 71.8 | 63.9 | 60.0 |
| | | LoRA | 67.0 | 17.4 | 42.4 | 41.1 | 38.6 | 70.1 | 64.2 | 55.4 |
| | | EWC | 78.1 | 75.5 | 69.8 | 53.1 | 62.6 | 77.6 | 72.0 | 66.3 |
| | | LwF | 99.6 | 99.8 | 99.6 | 97.6 | 99.8 | 99.3 | 99.4 | 100.0 |
| | | LP-FT | **99.8** | **100.0** | **99.8** | **97.9** | **100.0** | **99.8** | **99.8** | 100.0 |
| | | WiSE-FT | 88.6 | 72.2 | 78.1 | 70.3 | 73.4 | 88.2 | 80.6 | 91.7 |
| | | MS:PRE-FT-EWC-LwF | 90.4 | 86.6 | 84.6 | 76.6 | 79.7 | 89.8 | 85.9 | 90.5 |
| | IN-21K + AugReg | FT | 99.9 | **100.0** | 99.8 | 97.9 | 100.0 | 99.7 | 99.7 | 100.0 |
| | | Linear Probing | 84.0 | 67.6 | 73.9 | 75.0 | 73.2 | 84.0 | 69.5 | 88.8 |
| | | Visual Prompt | 69.3 | 40.6 | 63.8 | 49.4 | 56.5 | 74.5 | 65.3 | 62.8 |
| | | LoRA | 70.7 | 29.4 | 49.8 | 45.9 | 45.1 | 71.5 | 65.6 | 16.2 |
| | | EWC | 79.5 | 84.9 | 75.2 | 57.1 | 68.8 | 79.4 | 73.6 | 68.9 |
| | | LwF | 99.7 | 99.9 | 99.7 | 97.6 | 99.9 | 99.4 | 99.3 | 100.0 |
| | | LP-FT | **99.9** | **100.0** | **99.8** | **98.0** | **100.0** | **99.9** | **99.9** | **100.0** |
| | | WiSE-FT | 90.2 | 82.7 | 82.9 | 74.9 | 78.3 | 89.6 | 81.2 | 90.4 |
| | | MS:PRE-FT-EWC-LwF | 91.0 | 91.8 | 87.4 | 79.1 | 85.3 | 90.7 | 85.6 | 92.7 |
| ViT-S/32 | IN-21K + AugReg | FT | 99.9 | **100.0** | 99.8 | 97.8 | 100.0 | 99.6 | 99.7 | 100.0 |
| | | Linear Probing | 78.3 | 50.0 | 68.0 | 68.0 | 63.9 | 79.7 | 64.1 | 83.4 |
| | | Visual Prompt | 60.7 | 21.4 | 54.1 | 40.4 | 43.8 | 66.6 | 57.0 | 54.7 |
| | | LoRA | 64.0 | 12.5 | 42.4 | 39.9 | 35.6 | 65.5 | 57.3 | 36.2 |
| | | EWC | 73.7 | 67.9 | 67.1 | 50.4 | 57.3 | 73.5 | 66.7 | 61.5 |
| | | LwF | 99.6 | 99.9 | 99.6 | 97.5 | 99.9 | 99.3 | 99.4 | 100.0 |
| | | LP-FT | **100.0** | **100.0** | **99.8** | **97.9** | **100.0** | **99.8** | **99.9** | **100.0** |
| | | WiSE-FT | 87.4 | 67.1 | 77.9 | 69.7 | 71.2 | 87.8 | 77.0 | 90.9 |
| | | MS:PRE-FT-EWC-LwF | 88.7 | 81.7 | 83.1 | 75.8 | 78.6 | 88.7 | 82.4 | 88.6 |
| ViT-L/16 | IN-21K + AugReg | FT | 99.9 | **100.0** | **99.8** | 98.0 | **100.0** | 99.9 | 99.9 | **100.0** |
| | | Linear Probing | 98.3 | 98.1 | 94.0 | 93.1 | 92.9 | 96.9 | 91.7 | 99.1 |
| | | Visual Prompt | 71.3 | 48.0 | 69.4 | 50.8 | 59.6 | 76.0 | 66.7 | 67.1 |
| | | LoRA | 76.5 | 59.9 | 66.4 | 54.7 | 56.8 | 80.0 | 68.6 | 72.7 |
| | | EWC | 82.9 | 91.2 | 87.9 | 70.9 | 87.0 | 85.1 | 82.8 | 80.5 |
| | | LwF | 99.9 | 100.0 | 99.8 | 97.8 | 100.0 | 99.8 | 99.8 | 100.0 |
| | | LP-FT | **100.0** | **100.0** | **99.8** | **98.1** | **100.0** | **99.9** | **99.9** | 99.8 |
| | | WiSE-FT | 93.1 | 77.7 | 93.5 | 85.7 | 88.3 | 93.8 | 90.3 | 96.1 |
| | | MS:PRE-FT-EWC-LwF | 93.3 | 90.5 | 92.9 | 87.8 | 93.0 | 93.9 | 90.9 | 97.1 |

Table 21: Training accuracy on downstream datasets after fine-tuning with each method. FT and LP-FT generally achieve the highest performance, while EWC shows the lowest.

| Arch. | Pretraining Dataset | Method | IN-V2 | IN-A | IN-R | IN-Sketch | ObjNet | IN-Cartoon | IN-Drawing | IN-C |
|---|---|---|---|---|---|---|---|---|---|---|
| ResNet-18 | IN-1K | FT | 98.3 | **98.7** | **99.1** | 95.7 | **97.6** | 97.4 | 97.1 | 100.0 |
| | | Linear Probing | 59.9 | 6.5 | 47.0 | 33.3 | 34.2 | 65.1 | 48.3 | 50.8 |
| | | Visual Prompt | 52.4 | 5.8 | 35.9 | 22.9 | 29.1 | 46.8 | 28.2 | 28.1 |
| | | EWC | 63.0 | 17.5 | 50.8 | 37.9 | 38.0 | 66.0 | 54.0 | 42.6 |
| | | LwF | 97.0 | 97.6 | 98.3 | 94.7 | 96.7 | 96.2 | 95.8 | 99.9 |
| | | LP-FT | **98.5** | 98.5 | 98.9 | **95.8** | 97.4 | **97.7** | **97.3** | **100.0** |
| | | WiSE-FT | 80.2 | 30.7 | 69.4 | 53.5 | 58.7 | 75.9 | 58.6 | 75.6 |
| | | MS:PRE-FT-EWC-LwF | 80.8 | 41.8 | 72.9 | 59.9 | 62.4 | 80.3 | 70.3 | 74.9 |
| ResNet-50 | IN-1K | FT | 95.3 | 94.5 | **98.6** | 96.2 | 96.9 | 97.4 | 97.8 | 100.0 |
| | | Linear Probing | 69.8 | 19.8 | 52.3 | 32.9 | 46.2 | 75.0 | 57.0 | 55.5 |
| | | Visual Prompt | 66.0 | 22.4 | 47.3 | 33.3 | 45.6 | 59.3 | 39.3 | 42.1 |
| | | EWC | 72.0 | 43.1 | 58.4 | 44.5 | 49.5 | 76.3 | 63.6 | 52.4 |
| | | LwF | 94.6 | 94.0 | 97.9 | 95.4 | 95.6 | 96.4 | 96.8 | 100.0 |
| | | LP-FT | **95.7** | **94.8** | 98.6 | **96.2** | **97.0** | **97.5** | **97.9** | **100.0** |
| | | WiSE-FT | 82.7 | 56.6 | 73.7 | 56.7 | 68.6 | 82.5 | 65.9 | 83.8 |
| | | MS:PRE-FT-EWC-LwF | 84.1 | 66.7 | 78.9 | 62.4 | 71.6 | 86.4 | 76.0 | 84.8 |

### H.4 Robustness Improvement Results of Different Models

Across the ImageNet pretrained models, WiSE-FT and Model Soup consistently have better robustness improvement compared to other methods fine-tuning on realistic OOD datasets (Tables 22-26). Linear Probing consistently achieves the best robustness improvement using LAION-2B pretrained models (Table 27) and OpenAI CLIP models (Table 28).

Table 22: *RI* and *mRI* of ImageNet-1K with AugReg pretrained models with different fine-tuning methods and downstream datasets on each dataset in ImageNet-RIB.

| Architecture | Method | *mRI* | RI | | | | | | | |
| --- | --- | --- | --- | --- | --- | --- | --- | --- | --- | --- |
| | | | IN-V2 | IN-A | IN-R | IN-Sketch | ObjNet | IN-Cartoon | IN-Drawing | IN-C |
| ViT-B/16 | FT | 1.3 | 2.9 | -4.0 | 2.8 | 4.4 | -2.7 | 0.6 | 0.4 | 5.9 |
| | Linear Probing | 0.7 | 0.1 | -0.1 | 0.8 | 1.2 | 0.3 | 0.2 | 0.1 | 3.2 |
| | Visual Prompt | -4.5 | -2.3 | -9.1 | -4.9 | -1.6 | -11.2 | -3.9 | -4.3 | 1.7 |
| | LoRA | 0.9 | 0.2 | 0.4 | 1.1 | 2.6 | 0.3 | -0.1 | 1.3 | 1.1 |
| | EWC | 2.8 | 2.9 | -0.2 | 5.2 | 4.4 | 1.4 | 1.6 | 2.8 | 4.3 |
| | LwF | 3.1 | 2.8 | -0.0 | 6.2 | 4.6 | 0.7 | 1.9 | 2.1 | 6.5 |
| | LP-FT | 2.3 | **3.0** | -0.9 | 5.2 | 4.5 | -0.1 | 1.2 | 0.6 | 4.7 |
| | WiSE-FT | 3.6 | 2.5 | **0.7** | 7.5 | 4.5 | 2.1 | 2.3 | 3.0 | 6.5 |
| | MS:PRE-FT-EWC-LwF | **3.9** | 2.7 | 0.7 | **7.8** | **5.0** | **2.2** | **2.4** | **3.3** | **6.7** |
| ViT-B/32 | FT | -0.0 | 1.6 | -5.5 | 0.2 | 2.6 | -5.4 | 0.3 | -0.3 | 6.4 |
| | Linear Probing | 1.1 | 0.1 | -0.1 | 0.9 | 1.3 | 0.4 | 1.0 | 1.1 | 3.8 |
| | Visual Prompt | -5.4 | -2.7 | -13.3 | -4.7 | -2.0 | -12.7 | -2.4 | -5.0 | -0.1 |
| | LoRA | 0.9 | 0.3 | **0.3** | 0.5 | 1.0 | **0.7** | 0.7 | 0.5 | 3.1 |
| | EWC | 1.3 | **1.9** | -2.9 | 3.2 | 2.6 | 0.1 | 1.2 | 2.0 | 2.6 |
| | LwF | 1.8 | 1.5 | -2.0 | 3.9 | 3.2 | -1.9 | 1.4 | 1.2 | **6.9** |
| | LP-FT | 1.5 | 1.5 | -1.7 | 3.4 | 2.9 | -1.9 | 1.0 | 0.3 | 6.4 |
| | WiSE-FT | 2.5 | 1.5 | 0.2 | 5.0 | 3.3 | 0.3 | 1.6 | 2.2 | 6.1 |
| | MS:PRE-FT-EWC-LwF | **2.5** | 1.7 | -0.5 | **5.1** | **3.5** | 0.2 | **1.8** | **2.4** | 6.0 |
| ViT-S/16 | FT | -3.2 | -0.0 | -8.2 | -2.9 | 0.3 | -9.7 | -2.4 | -5.3 | 2.9 |
| | Linear Probing | 0.3 | 0.1 | -0.5 | 0.9 | 1.4 | -0.2 | -0.1 | 0.6 | -0.1 |
| | Visual Prompt | -7.4 | -4.6 | -13.3 | -6.1 | -3.5 | -18.1 | -6.3 | -6.0 | -1.4 |
| | LoRA | 0.9 | 0.2 | 0.1 | 1.6 | 3.6 | -0.1 | -0.3 | **1.5** | 0.8 |
| | EWC | 1.6 | **2.6** | -2.2 | 4.2 | **5.5** | -1.9 | 0.6 | 0.9 | 2.7 |
| | LwF | 0.6 | 0.9 | -1.5 | 3.5 | 1.5 | -2.7 | 0.3 | -2.4 | 5.4 |
| | LP-FT | -1.2 | 0.9 | -4.0 | 0.1 | 1.8 | -5.8 | -1.2 | -4.2 | 2.8 |
| | WiSE-FT | 2.9 | 2.2 | **0.7** | 6.5 | 4.7 | 0.1 | 1.9 | 1.4 | 5.8 |
| | MS:PRE-FT-EWC-LwF | **3.0** | 2.2 | 0.3 | **6.7** | 5.3 | **0.1** | **1.9** | 1.3 | **6.0** |
| ResNet-18 | FT | -5.2 | -2.1 | -11.7 | -0.6 | -5.0 | -8.8 | -5.7 | -13.6 | 5.7 |
| | Linear Probing | -7.3 | -1.4 | -2.5 | -1.2 | -26.9 | -3.9 | -4.7 | -15.5 | -2.1 |
| | Visual Prompt | -8.3 | -4.3 | -18.3 | -7.5 | -6.9 | -12.9 | -6.1 | -7.8 | -2.8 |
| | EWC | -5.7 | -0.6 | -9.6 | 2.0 | -11.7 | -4.3 | -4.6 | -15.1 | -1.5 |
| | LwF | -1.9 | -0.9 | -5.5 | 2.6 | -2.7 | -4.7 | -1.4 | -9.0 | **6.7** |
| | LP-FT | -4.8 | -2.2 | -10.0 | 1.0 | -6.2 | -7.1 | -5.7 | -13.9 | 6.1 |
| | WiSE-FT | **0.7** | **-0.1** | **-1.5** | 4.3 | **2.4** | **-1.5** | **-0.7** | **-2.8** | 5.3 |
| | MS:PRE-FT-EWC-LwF | -0.1 | -0.2 | -2.7 | 4.2 | 1.9 | -1.9 | -1.2 | -5.7 | 4.9 |
| ResNet-50 | FT | -5.2 | -0.1 | -2.9 | 2.8 | -10.7 | -4.3 | -6.5 | -22.4 | 2.4 |
| | Linear Probing | -11.2 | -1.5 | -1.2 | -1.2 | -37.0 | -4.2 | -5.6 | -35.2 | -3.9 |
| | Visual Prompt | -6.5 | -5.9 | -7.8 | -6.0 | -5.9 | -9.1 | -6.2 | -6.0 | -5.1 |
| | EWC | -8.9 | -1.1 | -0.5 | 2.2 | -21.7 | -3.2 | -7.2 | -36.2 | -3.3 |
| | LwF | -5.8 | 0.5 | -2.2 | 3.6 | -12.3 | -3.0 | -4.9 | -31.5 | 3.2 |
| | LP-FT | -5.1 | -0.2 | -2.6 | 3.2 | -10.3 | -4.1 | -6.4 | -22.1 | 1.9 |
| | WiSE-FT | **1.2** | **0.7** | **0.9** | 6.1 | **1.2** | **-0.0** | **-0.6** | **-3.0** | 4.3 |
| | MS:PRE-FT-EWC-LwF | -0.5 | 0.5 | 0.6 | **6.1** | 0.2 | -0.6 | -2.1 | -13.1 | **4.6** |

Table 23: *RI* and *mRI* of ImageNet-1K with SAM pretrained models with different fine-tuning methods and downstream datasets on each dataset in ImageNet-RIB.

| Architecture | Method | $mRI$ | RI | | | | | | | |
| --- | --- | --- | --- | --- | --- | --- | --- | --- | --- | --- |
| | | | IN-V2 | IN-A | IN-R | IN-Sketch | ObjNet | IN-Cartoon | IN-Drawing | IN-C |
| ViT-B/16 | FT | 2.5 | **3.4** | -3.2 | 5.8 | 5.7 | -2.3 | 1.7 | 1.8 | 7.3 |
| | Linear Probing | 0.8 | 0.1 | 0.3 | 0.5 | 1.1 | 0.0 | 0.0 | 0.4 | 4.0 |
| | Visual Prompt | -6.1 | -4.1 | -12.4 | -6.4 | -6.3 | -10.6 | -3.4 | -4.9 | -0.2 |
| | LoRA | 0.9 | 0.1 | 0.4 | 0.8 | 1.1 | 0.3 | 0.1 | 0.5 | 3.6 |
| | EWC | 1.6 | 0.7 | 0.3 | 3.7 | 2.4 | 1.2 | 1.0 | 2.0 | 1.3 |
| | LwF | 3.5 | 3.2 | -1.3 | 6.9 | **5.7** | 0.2 | 2.3 | 2.2 | **8.7** |
| | LP-FT | 2.4 | 3.3 | -2.3 | 6.4 | 5.6 | -1.5 | 1.7 | 1.5 | 4.8 |
| | WiSE-FT | 3.6 | 2.0 | **1.9** | 7.1 | 4.3 | 1.5 | **2.4** | 2.8 | 6.5 |
| | MS:PRE-FT-EWC-LwF | **3.7** | 2.0 | 1.7 | **7.3** | 4.6 | **1.6** | 2.4 | **3.0** | 6.7 |
| ViT-B/32 | FT | 1.4 | **2.4** | -4.6 | 4.2 | 3.8 | -4.0 | 0.9 | 0.8 | 7.6 |
| | Linear Probing | 0.9 | 0.1 | 0.4 | 0.8 | 1.1 | 0.2 | 0.3 | 0.2 | 4.2 |
| | Visual Prompt | -5.9 | -2.5 | -17.1 | -5.3 | -5.0 | -12.6 | -1.9 | -2.6 | -0.1 |
| | LoRA | 0.8 | 0.1 | 0.6 | 1.0 | 1.0 | 0.5 | 0.3 | 0.3 | 2.9 |
| | EWC | 1.0 | 0.6 | -0.3 | 2.5 | 2.1 | 0.6 | 0.6 | 1.1 | 1.0 |
| | LwF | 2.4 | 2.3 | -2.3 | **5.5** | **4.0** | -1.3 | 1.5 | 1.4 | 8.3 |
| | LP-FT | 1.9 | 2.3 | -3.0 | 5.1 | 3.7 | -2.8 | 1.0 | 0.4 | **8.3** |
| | WiSE-FT | 2.6 | 1.5 | **1.1** | 5.2 | 3.1 | **0.7** | **1.7** | 2.0 | 5.6 |
| | MS:PRE-FT-EWC-LwF | **2.6** | 1.5 | 0.8 | 5.4 | 3.4 | 0.7 | 1.7 | **2.1** | 5.5 |

Table 24: *RI* and *mRI* of ImageNet-21K pretrained models with different fine-tuning methods and downstream datasets on each dataset in ImageNet-RIB.

| Architecture | Method | $mRI$ | RI | | | | | | | |
| --- | --- | --- | --- | --- | --- | --- | --- | --- | --- | --- |
| | | | IN-V2 | IN-A | IN-R | IN-Sketch | ObjNet | IN-Cartoon | IN-Drawing | IN-C |
| ViT-B/16 | FT | -0.1 | 1.5 | -2.8 | 0.3 | 2.7 | -4.0 | -1.2 | -1.8 | 4.2 |
| | Linear Probing | 0.4 | 0.3 | 0.4 | 0.2 | 0.0 | 0.8 | -0.0 | 1.0 | 0.5 |
| | Visual Prompt | -9.4 | -7.7 | -12.7 | -11.1 | -7.7 | -14.8 | -8.4 | -10.0 | -3.3 |
| | LoRA | -0.3 | 0.2 | 0.5 | -1.6 | -0.5 | **0.9** | -0.4 | 0.6 | -1.9 |
| | EWC | 1.4 | 1.5 | 0.2 | 2.4 | 2.9 | 0.1 | 0.5 | 1.2 | 2.6 |
| | LwF | 1.6 | 1.5 | -0.5 | 2.9 | 3.5 | -0.8 | 0.9 | 0.0 | 5.3 |
| | LP-FT | 0.5 | 1.6 | -1.2 | 2.2 | 2.7 | -1.8 | -0.5 | -1.3 | 2.1 |
| | WiSE-FT | 2.5 | 1.7 | **0.8** | **4.9** | 3.8 | 0.6 | 1.3 | 1.7 | 5.5 |
| | MS:PRE-FT-EWC-LwF | **2.7** | **1.7** | 0.7 | 4.7 | **4.2** | 0.6 | **1.4** | **1.8** | **6.0** |

Table 25: *RI* and *mRI* of ImageNet-21K with AugReg pretrained models with different fine-tuning methods and downstream datasets on each dataset in ImageNet-RIB.

| Architecture | Method | *mRI* | *RI* | | | | | | | |
|---|---|---|---|---|---|---|---|---|---|---|
| | | | IN-V2 | IN-A | IN-R | IN-Sketch | ObjNet | IN-Cartoon | IN-Drawing | IN-C |
| ViT-B/16 | FT | -5.5 | -1.4 | -9.1 | -5.4 | -5.3 | -11.1 | -3.6 | -5.7 | -2.5 |
| | Linear Probing | -0.3 | -0.4 | -0.4 | -0.9 | -1.3 | -0.5 | 0.4 | 0.6 | 0.2 |
| | Visual Prompt | -8.0 | -6.1 | -9.2 | -9.3 | -6.1 | -16.6 | -7.1 | -7.0 | -3.0 |
| | LoRA | -2.1 | 0.7 | **0.8** | 2.6 | 2.9 | **0.8** | 0.8 | 1.6 | -27.4 |
| | EWC | 0.6 | **2.0** | -2.0 | 2.3 | **3.5** | -3.5 | 0.4 | 0.5 | 1.7 |
| | LwF | -1.0 | -1.0 | -2.3 | 0.5 | -1.5 | -4.2 | 0.3 | -1.3 | 1.7 |
| | LP-FT | -2.6 | 0.3 | -3.5 | -4.7 | -3.0 | -6.2 | -0.6 | -2.5 | -0.5 |
| | WiSE-FT | 1.7 | 1.8 | -0.2 | 4.0 | 2.4 | -0.9 | 2.0 | 1.5 | 3.2 |
| | MS:PRE-FT-EWC-LwF | **2.2** | 1.9 | 0.3 | **4.6** | 2.8 | -0.7 | **2.1** | **1.7** | **5.0** |
| ViT-B/32 | FT | -0.1 | 0.7 | -3.9 | 0.8 | 2.7 | -4.9 | -0.1 | -0.6 | 4.4 |
| | Linear Probing | 0.3 | -0.1 | -0.8 | 0.1 | 0.7 | -0.0 | 0.7 | 1.2 | 0.7 |
| | Visual Prompt | -8.4 | -4.9 | -13.1 | -7.8 | -5.0 | -20.7 | -6.2 | -7.3 | -2.4 |
| | LoRA | 0.9 | 0.0 | 0.5 | 1.0 | 1.2 | 0.7 | 0.1 | 1.5 | 2.0 |
| | EWC | 1.6 | **1.9** | -0.7 | 4.0 | 3.9 | -1.0 | 1.0 | 1.7 | 2.3 |
| | LwF | 1.7 | 1.0 | -0.5 | 3.9 | 2.8 | -1.3 | 1.7 | 1.2 | 5.0 |
| | LP-FT | 1.2 | 1.1 | -1.3 | 3.3 | 2.1 | -0.9 | 1.5 | 0.8 | 3.0 |
| | WiSE-FT | **3.0** | 1.7 | **0.9** | 5.6 | 4.0 | **1.0** | **2.0** | **2.5** | **6.0** |
| | MS:PRE-FT-EWC-LwF | 2.8 | 1.7 | 0.6 | 5.6 | **4.1** | 0.6 | 2.0 | 2.4 | 5.6 |
| ViT-S/16 | FT | -2.3 | -0.2 | -5.4 | -0.8 | 0.4 | -8.5 | -1.8 | -4.1 | 1.8 |
| | Linear Probing | -0.2 | -0.1 | -0.8 | -0.1 | 0.3 | -0.3 | 0.3 | 0.6 | -1.2 |
| | Visual Prompt | -9.2 | -5.7 | -12.1 | -8.9 | -5.0 | -21.3 | -8.3 | -9.6 | -2.8 |
| | LoRA | -1.5 | 0.1 | 0.4 | 1.5 | 2.8 | **0.5** | 0.3 | 1.6 | -19.5 |
| | EWC | 1.6 | **2.0** | -0.8 | 4.2 | **4.8** | -1.7 | 0.8 | 1.0 | 2.7 |
| | LwF | 0.5 | 0.5 | -0.8 | 3.2 | 1.4 | -3.1 | 0.9 | -1.3 | 3.4 |
| | LP-FT | -0.8 | 0.5 | -2.9 | 1.6 | 1.1 | -4.6 | -0.5 | -2.3 | 0.7 |
| | WiSE-FT | 2.8 | 1.8 | **0.8** | 6.1 | 4.4 | -0.1 | 1.9 | **2.0** | 5.1 |
| | MS:PRE-FT-EWC-LwF | **2.8** | 1.7 | 0.6 | **6.3** | 4.6 | -0.2 | **2.0** | 1.8 | **5.9** |
| ViT-S/32 | FT | -2.9 | -1.2 | -8.1 | -1.3 | 0.1 | -9.3 | -2.5 | -4.9 | 4.2 |
| | Linear Probing | -0.1 | -0.1 | -1.5 | 0.1 | 0.6 | -0.2 | 0.5 | 0.1 | -0.2 |
| | Visual Prompt | -9.6 | -4.7 | -21.6 | -8.5 | -5.6 | -19.1 | -5.7 | -8.6 | -2.7 |
| | LoRA | 0.4 | 0.1 | **0.5** | 1.1 | 2.7 | **0.5** | 0.3 | 1.1 | -3.0 |
| | EWC | 1.0 | **1.5** | -3.1 | 3.6 | **4.2** | -1.5 | 0.2 | 0.7 | 2.2 |
| | LwF | 0.3 | -0.1 | -2.1 | 3.2 | 1.6 | -4.0 | 0.5 | -1.5 | 4.8 |
| | LP-FT | -1.1 | -0.5 | -4.5 | 1.5 | 0.8 | -5.1 | -0.9 | -3.0 | 3.0 |
| | WiSE-FT | **2.3** | 1.1 | 0.1 | 5.5 | 3.8 | -0.6 | 1.2 | **1.2** | **6.2** |
| | MS:PRE-FT-EWC-LwF | 2.3 | 1.1 | -0.5 | **5.6** | 4.1 | -0.6 | **1.2** | 1.2 | 6.0 |
| ViT-L/16 | FT | -2.1 | 0.3 | -8.7 | -3.5 | -0.6 | -3.1 | -0.8 | -0.9 | 0.1 |
| | Linear Probing | -1.3 | -0.5 | -4.1 | -6.1 | -1.2 | -0.5 | 0.7 | 0.7 | 0.7 |
| | Visual Prompt | -12.9 | -10.7 | -13.6 | -13.5 | -15.0 | -17.0 | -10.4 | -14.2 | -9.0 |
| | LoRA | 1.0 | 0.2 | 0.7 | 1.1 | 1.2 | 0.9 | 0.7 | 1.1 | 1.7 |
| | EWC | 1.1 | -0.6 | 0.3 | 2.5 | 2.3 | -0.7 | 1.2 | 1.5 | 1.9 |
| | LwF | -0.2 | -0.6 | 0.4 | -1.9 | 0.5 | -0.2 | -0.6 | -1.8 | 2.6 |
| | LP-FT | -3.5 | 0.5 | -14.0 | -16.4 | -0.5 | -0.8 | 1.3 | 0.8 | 0.8 |
| | WiSE-FT | 2.3 | **2.1** | 0.1 | 3.3 | **2.6** | 1.1 | **2.4** | 2.7 | 4.4 |
| | MS:PRE-FT-EWC-LwF | **2.5** | 1.8 | **1.1** | **3.4** | 2.5 | **1.1** | 2.0 | **2.7** | **5.1** |

Table 26: *RI* and *mRI* of ImageNet-21K-P pretrained models with different fine-tuning methods and downstream datasets on each dataset in ImageNet-RIB.

| Architecture | Method | *mRI* | *RI* | | | | | | | |
| --- | --- | --- | --- | --- | --- | --- | --- | --- | --- | --- |
| | | | IN-V2 | IN-A | IN-R | IN-Sketch | ObjNet | IN-Cartoon | IN-Drawing | IN-C |
| | FT | -0.5 | 0.7 | -3.5 | 1.6 | 3.0 | -4.4 | -1.4 | -2.2 | 2.3 |
| | Linear Probing | 0.2 | 0.2 | 0.4 | 0.5 | 1.1 | 0.3 | 0.2 | 0.3 | -1.0 |
| | Visual Prompt | -10.1 | -8.0 | -11.5 | -9.9 | -7.8 | -19.9 | -8.8 | -11.1 | -3.6 |
| | LoRA | 0.4 | 0.1 | 0.3 | 0.5 | 1.2 | 0.5 | -0.2 | 0.9 | -0.1 |
| ViT-B/16 | EWC | 1.3 | 1.3 | 0.6 | 1.1 | 3.0 | 0.8 | 0.6 | 0.6 | 2.1 |
| | LwF | 1.7 | 1.6 | -0.1 | 4.5 | 3.5 | -0.3 | 1.0 | -0.2 | 3.7 |
| | LP-FT | 0.4 | 0.8 | -1.1 | 3.8 | 3.3 | -1.2 | -0.4 | -1.8 | 0.1 |
| | WiSE-FT | **3.0** | 2.0 | **1.3** | 6.3 | 4.1 | **1.3** | **1.8** | 2.0 | **5.3** |
| | MS:PRE-FT-EWC-LwF | 3.0 | **2.0** | 1.1 | **6.3** | **4.3** | 1.3 | 1.7 | **2.0** | 5.3 |

Table 27: *RI* and *mRI* of LAION-2B pretrained models with different fine-tuning methods and downstream datasets on each dataset in ImageNet-RIB.

| Architecture | Method | *mRI* | *RI* | | | | | | | |
| --- | --- | --- | --- | --- | --- | --- | --- | --- | --- | --- |
| | | | IN-V2 | IN-A | IN-R | IN-Sketch | ObjNet | IN-Cartoon | IN-Drawing | IN-C |
| | FT | -38.1 | -39.4 | -53.7 | -36.9 | -46.9 | -49.2 | -29.0 | -36.6 | -12.9 |
| | Linear Probing | **-2.0** | **-0.6** | **-0.9** | **-1.5** | **-0.8** | **-1.9** | **-2.4** | **-5.5** | -2.0 |
| | Visual Prompt | -8.2 | -6.4 | -8.1 | -8.1 | -6.7 | -16.7 | -8.3 | -8.0 | -2.9 |
| | LoRA | -3.6 | -0.8 | -1.3 | -1.6 | -1.2 | -3.0 | -2.8 | -5.8 | -12.3 |
| ViT-B/16 | EWC | -12.5 | -16.1 | -27.2 | -2.4 | -17.3 | -19.0 | -6.4 | -9.9 | **-1.7** |
| | LwF | -33.9 | -37.3 | -49.3 | -31.7 | -45.2 | -44.7 | -22.3 | -31.0 | -9.9 |
| | LP-FT | -37.1 | -39.3 | -51.0 | -35.9 | -46.1 | -47.7 | -28.3 | -33.7 | -14.6 |
| | WiSE-FT | -21.6 | -25.3 | -39.1 | -17.6 | -31.9 | -25.4 | -11.3 | -16.3 | -5.5 |
| | MS:PRE-FT-EWC-LwF | -17.9 | -21.1 | -31.3 | -12.9 | -29.7 | -22.1 | -8.6 | -14.6 | -2.7 |
| | FT | -31.6 | -31.1 | -47.0 | -28.9 | -37.5 | -41.3 | -24.5 | -32.8 | -9.6 |
| | Linear Probing | **-1.4** | **-0.1** | **-1.5** | **0.2** | **0.5** | **-2.1** | **-2.3** | **-6.0** | 0.5 |
| | Visual Prompt | -8.4 | -6.4 | -12.1 | -6.9 | -6.0 | -21.9 | -6.1 | -6.7 | -1.5 |
| | LoRA | -1.9 | -0.2 | -2.0 | -0.4 | -0.9 | -4.0 | -2.6 | -6.1 | **1.1** |
| ViT-B/32 | EWC | -10.0 | -10.6 | -25.6 | -1.2 | -11.5 | -15.1 | -3.5 | -11.0 | -1.1 |
| | LwF | -26.7 | -28.5 | -40.5 | -22.8 | -33.7 | -34.4 | -18.5 | -26.8 | -8.6 |
| | LP-FT | -30.8 | -31.3 | -45.7 | -27.9 | -35.9 | -39.7 | -24.4 | -30.8 | -10.3 |
| | WiSE-FT | -13.5 | -15.5 | -22.8 | -8.6 | -17.8 | -17.9 | -8.4 | -14.4 | -2.2 |
| | MS:PRE-FT-EWC-LwF | -10.9 | -12.4 | -19.0 | -5.7 | -16.4 | -14.3 | -5.7 | -12.5 | -1.3 |

Table 28: *RI* and *mRI* of OpenAI CLIP models with different fine-tuning methods and downstream datasets on each dataset in ImageNet-RIB.

| Architecture | Method | *mRI* | *RI* | | | | | | | |
| --- | --- | --- | --- | --- | --- | --- | --- | --- | --- | --- |
| | | | IN-V2 | IN-A | IN-R | IN-Sketch | ObjNet | IN-Cartoon | IN-Drawing | IN-C |
| | FT | -38.0 | -38.3 | -51.6 | -35.4 | -48.5 | -50.3 | -28.9 | -35.8 | -15.3 |
| | Linear Probing | **-2.0** | **-0.5** | **-0.8** | **-1.3** | -1.3 | **-1.2** | **-3.4** | **-5.6** | -1.8 |
| | Visual Prompt | -8.4 | -7.4 | -8.1 | -7.6 | -6.3 | -16.3 | -9.4 | -9.9 | -2.7 |
| | LoRA | -3.6 | -0.6 | -1.0 | -1.9 | **-1.0** | -2.8 | -4.0 | -6.4 | -11.3 |
| ViT-B/16 | EWC | -12.7 | -14.4 | -20.9 | -2.4 | -24.8 | -19.9 | -7.5 | -10.8 | **-0.8** |
| | LwF | -33.1 | -35.5 | -46.4 | -30.6 | -47.1 | -44.3 | -22.7 | -30.2 | -7.9 |
| | LP-FT | -36.9 | -38.3 | -50.0 | -34.4 | -48.5 | -49.0 | -29.8 | -31.7 | -13.3 |
| | WiSE-FT | -18.1 | -19.5 | -26.7 | -11.7 | -31.0 | -23.7 | -11.1 | -15.8 | -5.5 |
| | MS:PRE-FT-EWC-LwF | -16.0 | -17.1 | -24.3 | -9.4 | -30.3 | -20.9 | -9.1 | -14.4 | -2.7 |
| | FT | -28.7 | -28.1 | -43.8 | -26.4 | -35.0 | -39.1 | -20.8 | -28.2 | -8.4 |
| | Linear Probing | **-1.3** | **0.2** | **-0.9** | -0.8 | **-0.1** | **-1.8** | **-2.1** | -5.6 | 0.9 |
| | Visual Prompt | -8.0 | -5.4 | -12.5 | -6.2 | -4.6 | -20.8 | -5.9 | -7.0 | -1.4 |
| | LoRA | -1.8 | 0.1 | -1.6 | **-0.8** | -0.6 | -3.7 | -2.3 | **-5.4** | -0.2 |
| ViT-B/32 | EWC | -7.0 | -5.6 | -17.0 | -1.1 | -11.4 | -13.0 | -3.1 | -6.5 | **1.7** |
| | LwF | -23.9 | -24.8 | -37.0 | -21.1 | -31.3 | -31.7 | -16.5 | -24.3 | -4.4 |
| | LP-FT | -27.7 | -27.4 | -42.2 | -24.3 | -33.7 | -37.7 | -20.2 | -26.9 | -9.0 |
| | WiSE-FT | -9.7 | -10.3 | -16.5 | -5.3 | -14.2 | -12.5 | -5.7 | -11.3 | -1.5 |
| | MS:PRE-FT-EWC-LwF | -8.1 | -8.5 | -14.7 | -3.2 | -13.4 | -10.6 | -4.4 | -9.9 | 0.5 |

**H.5 Accuracy of Using Various pretrained Models on Each OOD Datasets and Each Corruption in ImageNet-C**

Table 29 summarizes the Table indices for the accuracy on each OOD (out-of-distribution) dataset (Table S1 and Tables S17 and ImageNet-C (Table S18-Table S35) after fine-tuning on various datasets in the Supplementary Materials. Each pretrained and fine-tuned model is evaluated on ImageNet-C with 15 corruptions at severity levels ranging from 1 to 5. Following the original ImageNet-C benchmark (Hendrycks & Dietterich, 2019), we average the performance over the different severity levels. However, for consistency with other datasets, we report the results as accuracy rather than error.

Table 29: Reference for the tables showing accuracy of pretrained models on OOD datasets (left) and ImageNet-C corruptions (right).

| Architecture | $D_{\text{pre}}$ | Accuracy on OOD datasets | Accuracy on ImageNet-C |
|---|---|---|---|
| ViT-B/16 | IN-1K + AugReg | Table 30 (Table S1) | Table 31 (Table S2) |
| | IN-1K + SAM | Table S2 | Table S20 |
| | IN-21K | Table S3 | Table S21 |
| | IN-21K-P | Table S4 | Table S22 |
| | IN-21K + AugReg | Table S5 | Table S23 |
| | LAION-2B | Table S6 | Table S24 |
| | OpenAI | Table S7 | Table S25 |
| ViT-B/32 | IN-1K + AugReg | Table S8 | Table S26 |
| | IN-21K + AugReg | Table S9 | Table S27 |
| | LAION-2B | Table S10 | Table S28 |
| | OpenAI | Table S11 | Table S29 |
| ViT-S/16 | IN-1K + AugReg | Table S12 | Table S30 |
| | IN-21K + AugReg | Table S13 | Table S31 |
| ViT-S/32 | IN-21K + AugReg | Table S14 | Table S32 |
| ViT-L/16 | IN-21K + AugReg | Table S15 | Table S33 |
| ResNet-18 | IN-1K | Table S16 | Table S34 |
| ResNet-50 | IN-1K | Table S17 | Table S35 |

Table 30: The accuracy on each OOD dataset after fine-tuning on ImageNet-1K with AugReg pretrained ViT-B/16 on the downstream datasets with various methods. Note that ImageNet-Drawing, ImageNet-Cartoon, and ImageNet-C are generated from the ImageNet validation set. Green and red indicate relative performance increases and decreases, respectively, compared to the pretrained model. Bold indicates the best performance on each evaluation dataset.

| Method | Fine-Tuning Dataset | IN-1K | IN-V2 | IN-A | IN-R | IN-Sketch | ObjNet | IN-Cartoon | IN-Drawing | IN-C |
|---|---|---|---|---|---|---|---|---|---|---|
| Pretrained | | 79.2 | 66.4 | 15.0 | 38.0 | 28.0 | 25.7 | 66.2 | 39.1 | 56.0 |
| FT | IN-V2 | 78.4 | - | 25.2 | 41.9 | 29.2 | 37.1 | 64.7 | 40.4 | 57.4 |
| | IN-A | 72.9 | 60.6 | - | 36.7 | 24.9 | 35.0 | 55.3 | 32.6 | 53.5 |
| | IN-R | 69.8 | 59.2 | 20.9 | - | 46.7 | 32.0 | 61.3 | 51.4 | 52.0 |
| | IN-Sketch | 75.7 | 63.9 | 17.3 | 59.1 | - | 33.0 | 66.3 | 50.8 | 53.8 |
| | ObjNet | 74.4 | 62.2 | 24.9 | 36.3 | 25.1 | - | 55.6 | 33.6 | 52.3 |
| | IN-Cartoon | 85.2 | 63.5 | 19.9 | 40.5 | 29.5 | 33.5 | - | 41.2 | 51.3 |
| | IN-Drawing | 81.5 | 62.9 | 16.5 | 41.1 | 32.7 | 32.4 | 64.2 | - | 56.0 |
| | IN-C | 99.8 | 61.1 | 13.9 | 37.0 | 25.1 | 27.7 | 92.2 | 70.2 | - |
| Linear Probing | IN-V2 | 79.1 | - | 15.6 | 38.2 | 28.1 | 33.1 | 66.2 | 39.0 | 55.9 |
| | IN-A | 78.6 | 65.9 | - | 38.5 | 27.4 | 34.1 | 65.6 | 38.6 | 55.8 |
| | IN-R | 78.7 | 66.6 | 17.1 | - | 30.2 | 33.4 | 66.1 | 39.8 | 56.2 |
| | IN-Sketch | 77.2 | 64.8 | 16.6 | 46.3 | - | 33.5 | 65.6 | 40.5 | 54.5 |
| | ObjNet | 78.6 | 65.9 | 18.1 | 38.6 | 27.9 | - | 65.1 | 39.3 | 56.1 |
| | IN-Cartoon | 80.5 | 65.4 | 15.1 | 39.2 | 28.1 | 32.2 | - | 40.9 | 55.6 |
| | IN-Drawing | 78.1 | 65.2 | 14.9 | 41.3 | 28.5 | 33.3 | 65.6 | - | 54.3 |
| | IN-C | 97.1 | 61.9 | 15.1 | 36.8 | 25.2 | 28.3 | 83.3 | 57.4 | - |
| Visual Prompt (Bahng et al., 2022) | IN-V2 | 75.7 | - | 12.7 | 39.6 | 27.4 | 34.4 | 60.5 | 36.7 | 47.9 |
| | IN-A | 69.1 | 57.1 | - | 36.3 | 21.9 | 32.7 | 50.6 | 26.1 | 38.0 |
| | IN-R | 68.1 | 55.9 | 9.6 | - | 36.2 | 30.0 | 55.7 | 41.8 | 40.1 |
| | IN-Sketch | 72.2 | 59.5 | 9.4 | 51.6 | - | 32.3 | 60.6 | 44.9 | 44.3 |
| | ObjNet | 68.6 | 56.2 | 13.0 | 33.7 | 22.2 | - | 46.8 | 23.0 | 35.3 |
| | IN-Cartoon | 74.5 | 61.2 | 10.2 | 41.2 | 27.0 | 31.5 | - | 35.2 | 41.8 |
| | IN-Drawing | 72.1 | 59.4 | 8.4 | 42.2 | 28.8 | 30.6 | 59.3 | - | 44.2 |
| | IN-C | 77.9 | 65.2 | 14.8 | 40.1 | 28.3 | 35.7 | 63.5 | 49.8 | - |
| LoRA (Hu et al., 2021) | IN-V2 | 79.2 | - | 15.3 | 38.2 | 28.1 | 33.2 | 66.4 | 39.3 | 56.1 |
| | IN-A | 79.0 | 66.4 | - | 38.9 | 27.8 | 35.5 | 65.2 | 39.3 | 56.5 |
| | IN-R | 79.2 | 66.8 | 16.7 | - | 29.7 | 34.8 | 66.9 | 40.0 | 56.7 |
| | IN-Sketch | 79.2 | 66.8 | 16.5 | 45.9 | - | 34.6 | 67.7 | 44.1 | 56.6 |
| | ObjNet | 78.9 | 66.3 | 18.3 | 39.3 | 27.8 | - | 65.1 | 39.2 | 55.0 |
| | IN-Cartoon | 78.7 | 65.8 | 14.8 | 39.3 | 28.3 | 32.1 | - | 39.8 | 54.6 |
| | IN-Drawing | 77.9 | 66.3 | 15.0 | 43.7 | 32.1 | 33.5 | 66.4 | - | 55.1 |
| | IN-C | 79.9 | 67.4 | 16.3 | 39.2 | 28.1 | 34.1 | 67.5 | 40.8 | - |
| EWC (Kirkpatrick et al., 2017) | IN-V2 | 80.0 | - | 19.7 | 41.8 | 29.4 | 36.8 | 67.1 | 42.8 | 58.2 |
| | IN-A | 76.9 | 64.9 | - | 40.4 | 27.8 | 38.2 | 61.1 | 36.5 | 56.6 |
| | IN-R | 75.2 | 63.9 | 19.0 | - | 43.9 | 33.3 | 66.4 | 57.5 | 56.1 |
| | IN-Sketch | 78.9 | 66.6 | 16.6 | 52.2 | - | 34.2 | 68.3 | 49.6 | 57.2 |
| | ObjNet | 78.1 | 66.2 | 23.1 | 40.9 | 29.0 | - | 62.4 | 39.8 | 56.9 |
| | IN-Cartoon | 79.2 | 66.0 | 16.5 | 42.7 | 29.9 | 33.8 | - | 42.6 | 54.7 |
| | IN-Drawing | 79.3 | 66.7 | 16.3 | 44.5 | 34.0 | 34.7 | 67.9 | - | 58.3 |
| | IN-C | 80.1 | 67.8 | 20.0 | 42.5 | 31.2 | 37.5 | 66.8 | 50.0 | - |
| LwF (Li & Hoiem, 2017) | IN-V2 | 79.2 | - | 22.9 | 41.3 | 29.4 | 36.4 | 65.8 | 41.0 | 57.9 |
| | IN-A | 77.4 | 65.5 | - | 39.4 | 27.5 | 36.7 | 61.8 | 38.3 | 57.2 |
| | IN-R | 76.1 | 64.7 | 21.7 | - | 47.8 | 34.1 | 66.8 | 54.9 | 57.2 |
| | IN-Sketch | 77.3 | 65.2 | 17.3 | 57.8 | - | 33.5 | 67.8 | 49.6 | 55.2 |
| | ObjNet | 78.2 | 66.2 | 24.1 | 38.4 | 27.3 | - | 62.3 | 38.8 | 56.3 |
| | IN-Cartoon | 87.2 | 65.9 | 19.4 | 41.2 | 29.9 | 34.2 | - | 42.7 | 55.6 |
| | IN-Drawing | 84.0 | 65.4 | 17.7 | 41.9 | 33.2 | 33.4 | 67.7 | - | 58.2 |
| | IN-C | 99.2 | 65.8 | 13.5 | 40.7 | 27.8 | 31.4 | 90.6 | 61.7 | - |
| LP-FT (Kumar et al., 2022) | IN-V2 | 78.8 | - | 24.7 | 41.6 | 29.3 | 36.8 | 65.3 | 41.3 | 57.6 |
| | IN-A | 76.5 | 64.6 | - | 38.2 | 27.4 | 37.1 | 60.5 | 36.7 | 56.2 |
| | IN-R | 74.7 | 63.4 | 21.1 | - | 46.9 | 34.7 | 65.4 | 53.1 | 55.3 |
| | IN-Sketch | 76.2 | 64.5 | 18.0 | 58.8 | - | 33.9 | 67.0 | 48.9 | 54.4 |
| | ObjNet | 77.1 | 64.9 | 24.9 | 38.2 | 26.8 | - | 60.7 | 37.7 | 54.9 |
| | IN-Cartoon | 86.3 | 64.2 | 19.5 | 41.0 | 29.9 | 33.5 | - | 43.1 | 52.8 |
| | IN-Drawing | 82.1 | 63.2 | 16.5 | 41.7 | 32.9 | 32.0 | 64.8 | - | 56.0 |
| | IN-C | 98.0 | 61.0 | 13.7 | 37.5 | 25.7 | 27.3 | 87.1 | 66.0 | - |
| WiSE-FT (Wortsman et al., 2022b) | IN-V2 | 78.8 | - | 24.7 | 41.6 | 29.3 | 36.8 | 65.3 | 41.3 | 57.6 |
| | IN-V2 | 79.7 | - | 21.3 | 40.5 | 29.5 | 36.0 | 66.5 | 40.9 | 58.0 |
| | IN-A | 78.6 | 66.4 | - | 39.3 | 28.5 | 37.1 | 64.4 | 38.6 | 57.8 |
| | IN-R | 79.1 | 67.1 | 23.0 | - | 44.7 | 37.4 | 69.5 | 54.7 | 59.6 |
| | IN-Sketch | 78.9 | 66.4 | 17.6 | 52.1 | - | 34.7 | 68.7 | 48.7 | 57.3 |
| | ObjNet | 79.3 | 67.3 | 23.5 | 40.0 | 29.0 | - | 65.2 | 40.5 | 57.6 |
| | IN-Cartoon | 83.8 | 66.5 | 19.3 | 41.0 | 30.4 | 34.9 | - | 43.2 | 56.3 |
| | IN-Drawing | 82.5 | 66.9 | 18.5 | 42.2 | 33.5 | 35.0 | 68.2 | - | 59.5 |
| | IN-C | 93.4 | 66.9 | 18.7 | 41.3 | 29.9 | 34.7 | 82.4 | 57.6 | - |
| Model Soup PRE-FT-EWC-LwF (Wortsman et al., 2022a) | IN-V2 | 79.8 | - | 21.0 | 41.0 | 29.7 | 36.0 | 66.9 | 41.7 | 58.0 |
| | IN-A | 78.3 | 66.4 | - | 39.7 | 28.5 | 37.5 | 63.7 | 38.4 | 57.8 |
| | IN-R | 78.9 | 67.1 | 23.1 | - | 45.9 | 37.2 | 69.6 | 55.8 | 59.6 |
| | IN-Sketch | 78.9 | 66.6 | 17.5 | 54.0 | - | 34.6 | 69.1 | 49.8 | 57.5 |
| | ObjNet | 79.3 | 67.4 | 24.1 | 40.3 | 29.1 | - | 64.9 | 40.6 | 57.7 |
| | IN-Cartoon | 83.7 | 66.4 | 18.9 | 41.8 | 30.6 | 34.7 | - | 43.6 | 56.2 |
| | IN-Drawing | 82.6 | 66.9 | 18.4 | 43.0 | 34.0 | 35.2 | 68.7 | - | 59.7 |
| | IN-C | 92.6 | 67.5 | 18.6 | 42.3 | 30.6 | 35.3 | 81.3 | 57.3 | - |

Table 31: Accuracy of ImageNet-1K with AugReg pretrained ViT-B/16 with different fine-tuning methods and downstream datasets on each ImageNet-C corruption. For each corruption, accuracy is averaged across 5 levels of severity.

| Method | Fine-Tuning Dataset | Avg. | Noise | | | Blur | | | | Weather | | | | Digital | | | |
|---|---|---|---|---|---|---|---|---|---|---|---|---|---|---|---|---|---|
| | | | Gauss. | Shot | Impulse | Defocus | Glass | Motion | Zoom | Snow | Frost | Fog | Bright | Contrast | Elastic | Pixel | JPEG |
| Pretrained | | 56.0 | 57 | 54 | 54 | 49 | 42 | 53 | 46 | 48 | 55 | 61 | 74 | 56 | 59 | 67 | 66 |
| FT | IN-V2 | 57.4 | 56 | 54 | 53 | 51 | 40 | 55 | 46 | 53 | 59 | 65 | 74 | 59 | 58 | 68 | 67 |
| | IN-A | 53.5 | 53 | 51 | 50 | 50 | 38 | 52 | 39 | 50 | 56 | 57 | 70 | 56 | 51 | 65 | 64 |
| | IN-R | 52.0 | 52 | 50 | 49 | 46 | 44 | 49 | 37 | 49 | 55 | 57 | 66 | 53 | 51 | 62 | 61 |
| | IN-Sketch | 53.8 | 55 | 53 | 52 | 46 | 39 | 49 | 43 | 51 | 56 | 58 | 70 | 55 | 55 | 63 | 62 |
| | ObjNet | 52.3 | 52 | 48 | 48 | 46 | 37 | 51 | 38 | 50 | 55 | 58 | 70 | 51 | 52 | 64 | 63 |
| | IN-Cartoon | 51.3 | 53 | 50 | 50 | 44 | 35 | 48 | 35 | 48 | 50 | 54 | 74 | 53 | 53 | 65 | 58 |
| | IN-Drawing | 56.0 | 58 | 56 | 55 | 46 | 43 | 52 | 40 | 55 | 62 | 61 | 74 | 53 | 57 | 66 | 62 |
| Linear Probing | IN-V2 | 55.9 | 56 | 54 | 54 | 49 | 42 | 53 | 46 | 48 | 55 | 61 | 73 | 56 | 59 | 66 | 65 |
| | IN-A | 55.8 | 56 | 53 | 53 | 49 | 42 | 54 | 46 | 48 | 55 | 61 | 73 | 57 | 59 | 66 | 65 |
| | IN-R | 56.2 | 56 | 54 | 54 | 49 | 44 | 54 | 47 | 49 | 55 | 61 | 73 | 56 | 60 | 66 | 66 |
| | IN-Sketch | 54.5 | 54 | 52 | 52 | 48 | 41 | 51 | 45 | 48 | 54 | 59 | 72 | 55 | 58 | 65 | 64 |
| | ObjNet | 56.1 | 56 | 54 | 54 | 49 | 43 | 54 | 48 | 48 | 56 | 62 | 73 | 53 | 60 | 66 | 65 |
| | IN-Cartoon | 55.6 | 56 | 54 | 53 | 48 | 42 | 52 | 46 | 48 | 55 | 59 | 75 | 54 | 59 | 67 | 67 |
| | IN-Drawing | 54.3 | 57 | 55 | 55 | 43 | 43 | 50 | 44 | 51 | 61 | 49 | 74 | 39 | 59 | 66 | 67 |
| Visual Prompt (Bahng et al., 2022) | IN-V2 | 47.9 | 44 | 42 | 41 | 41 | 35 | 46 | 42 | 42 | 46 | 51 | 69 | 48 | 55 | 59 | 57 |
| | IN-A | 38.0 | 33 | 31 | 29 | 31 | 24 | 36 | 31 | 35 | 38 | 43 | 60 | 37 | 46 | 48 | 49 |
| | IN-R | 40.1 | 39 | 38 | 36 | 33 | 28 | 36 | 30 | 36 | 41 | 41 | 61 | 38 | 45 | 50 | 50 |
| | IN-Sketch | 44.3 | 43 | 41 | 40 | 37 | 29 | 40 | 36 | 39 | 45 | 46 | 65 | 47 | 49 | 54 | 55 |
| | ObjNet | 35.3 | 28 | 26 | 24 | 28 | 22 | 33 | 29 | 32 | 35 | 41 | 61 | 37 | 44 | 45 | 44 |
| | IN-Cartoon | 41.8 | 39 | 37 | 36 | 34 | 27 | 38 | 33 | 36 | 38 | 42 | 66 | 43 | 50 | 55 | 53 |
| | IN-Drawing | 44.2 | 45 | 43 | 43 | 33 | 32 | 38 | 32 | 41 | 51 | 42 | 65 | 39 | 50 | 56 | 52 |
| LoRA (Hu et al., 2021) | IN-V2 | 56.1 | 57 | 54 | 54 | 49 | 43 | 53 | 46 | 48 | 55 | 61 | 74 | 57 | 59 | 67 | 66 |
| | IN-A | 56.5 | 57 | 54 | 54 | 49 | 44 | 55 | 48 | 49 | 57 | 61 | 74 | 52 | 60 | 67 | 66 |
| | IN-R | 56.7 | 57 | 54 | 54 | 50 | 44 | 54 | 48 | 49 | 56 | 62 | 74 | 56 | 60 | 67 | 66 |
| | IN-Sketch | 56.6 | 56 | 54 | 54 | 51 | 43 | 53 | 47 | 50 | 56 | 62 | 74 | 57 | 59 | 67 | 66 |
| | ObjNet | 55.0 | 57 | 54 | 54 | 48 | 43 | 54 | 47 | 48 | 55 | 55 | 74 | 44 | 60 | 67 | 66 |
| | IN-Cartoon | 54.6 | 56 | 53 | 53 | 48 | 43 | 50 | 45 | 48 | 54 | 56 | 73 | 50 | 58 | 66 | 65 |
| | IN-Drawing | 55.1 | 58 | 56 | 56 | 44 | 45 | 51 | 43 | 51 | 63 | 54 | 74 | 43 | 59 | 66 | 66 |
| EWC (Kirkpatrick et al., 2017) | IN-V2 | 58.2 | 58 | 55 | 55 | 52 | 44 | 56 | 49 | 52 | 58 | 64 | 75 | 59 | 61 | 68 | 67 |
| | IN-A | 56.6 | 55 | 53 | 52 | 52 | 42 | 56 | 46 | 52 | 58 | 62 | 73 | 59 | 57 | 67 | 66 |
| | IN-R | 56.1 | 55 | 54 | 53 | 50 | 44 | 53 | 43 | 53 | 59 | 62 | 72 | 58 | 56 | 64 | 65 |
| | IN-Sketch | 57.2 | 57 | 56 | 55 | 50 | 44 | 54 | 47 | 52 | 57 | 61 | 74 | 57 | 59 | 67 | 67 |
| | ObjNet | 56.9 | 56 | 53 | 53 | 51 | 43 | 56 | 47 | 52 | 58 | 62 | 74 | 58 | 59 | 67 | 66 |
| | IN-Cartoon | 54.7 | 55 | 52 | 52 | 48 | 40 | 52 | 43 | 48 | 54 | 60 | 73 | 56 | 58 | 66 | 64 |
| | IN-Drawing | 58.3 | 59 | 57 | 57 | 50 | 44 | 55 | 45 | 54 | 63 | 65 | 74 | 59 | 60 | 68 | 66 |
| LwF (Li & Hoiem, 2017) | IN-V2 | 57.9 | 57 | 55 | 54 | 51 | 42 | 55 | 47 | 53 | 59 | 65 | 75 | 60 | 59 | 69 | 68 |
| | IN-A | 57.2 | 56 | 54 | 54 | 52 | 42 | 55 | 45 | 53 | 60 | 62 | 73 | 59 | 57 | 68 | 66 |
| | IN-R | 57.2 | 57 | 56 | 55 | 50 | 48 | 54 | 43 | 54 | 59 | 62 | 72 | 57 | 57 | 67 | 66 |
| | IN-Sketch | 55.2 | 56 | 54 | 53 | 48 | 40 | 51 | 45 | 52 | 57 | 60 | 72 | 56 | 57 | 65 | 64 |
| | ObjNet | 56.3 | 56 | 53 | 53 | 51 | 41 | 55 | 44 | 52 | 57 | 63 | 73 | 57 | 57 | 67 | 66 |
| | IN-Cartoon | 55.6 | 56 | 53 | 53 | 49 | 40 | 52 | 41 | 51 | 55 | 59 | 77 | 57 | 58 | 68 | 65 |
| | IN-Drawing | 58.2 | 59 | 56 | 56 | 50 | 45 | 55 | 43 | 55 | 63 | 64 | 77 | 56 | 59 | 69 | 65 |
| LP-FT (Kumar et al., 2022) | IN-V2 | 57.6 | 57 | 54 | 54 | 51 | 41 | 55 | 46 | 53 | 59 | 65 | 74 | 60 | 59 | 68 | 67 |
| | IN-A | 56.2 | 55 | 52 | 52 | 51 | 41 | 55 | 43 | 53 | 59 | 62 | 73 | 59 | 56 | 67 | 65 |
| | IN-R | 55.3 | 55 | 54 | 52 | 48 | 47 | 52 | 41 | 52 | 58 | 60 | 70 | 56 | 56 | 65 | 64 |
| | IN-Sketch | 54.4 | 54 | 53 | 52 | 48 | 40 | 50 | 44 | 51 | 55 | 59 | 70 | 56 | 56 | 64 | 63 |
| | ObjNet | 54.9 | 54 | 51 | 51 | 48 | 40 | 54 | 43 | 51 | 57 | 61 | 72 | 54 | 56 | 66 | 64 |
| | IN-Cartoon | 52.8 | 53 | 50 | 50 | 46 | 37 | 49 | 38 | 49 | 52 | 55 | 75 | 54 | 55 | 66 | 61 |
| | IN-Drawing | 56.0 | 59 | 56 | 56 | 44 | 44 | 52 | 40 | 56 | 63 | 57 | 76 | 49 | 58 | 67 | 64 |
| WiSE-FT (Wortsman et al., 2022b) | IN-V2 | 58.0 | 58 | 55 | 55 | 51 | 42 | 55 | 47 | 52 | 58 | 65 | 75 | 60 | 60 | 69 | 68 |
| | IN-A | 57.8 | 57 | 55 | 55 | 52 | 43 | 56 | 46 | 53 | 59 | 64 | 74 | 60 | 59 | 68 | 66 |
| | IN-R | 59.6 | 59 | 58 | 57 | 53 | 49 | 57 | 48 | 55 | 61 | 65 | 75 | 60 | 61 | 69 | 68 |
| | IN-Sketch | 57.3 | 58 | 56 | 56 | 50 | 42 | 53 | 47 | 53 | 59 | 63 | 74 | 59 | 59 | 67 | 66 |
| | ObjNet | 57.6 | 57 | 54 | 54 | 51 | 43 | 56 | 46 | 53 | 58 | 64 | 74 | 59 | 59 | 68 | 67 |
| | IN-Cartoon | 56.3 | 57 | 54 | 55 | 50 | 41 | 53 | 43 | 51 | 55 | 61 | 76 | 58 | 59 | 68 | 65 |
| | IN-Drawing | 59.5 | 61 | 59 | 59 | 51 | 45 | 56 | 46 | 55 | 63 | 65 | 77 | 59 | 61 | 69 | 67 |
| Model Soup PRE-FT-EWC-LwF (Wortsman et al., 2022a) | IN-V2 | 58.0 | 58 | 55 | 55 | 51 | 43 | 55 | 47 | 52 | 59 | 64 | 75 | 60 | 60 | 69 | 68 |
| | IN-A | 57.8 | 57 | 55 | 54 | 52 | 43 | 56 | 46 | 53 | 59 | 64 | 74 | 60 | 58 | 68 | 67 |
| | IN-R | 59.6 | 59 | 58 | 57 | 53 | 49 | 57 | 47 | 55 | 61 | 65 | 74 | 60 | 61 | 69 | 68 |
| | IN-Sketch | 57.5 | 58 | 56 | 56 | 50 | 42 | 53 | 47 | 53 | 59 | 63 | 74 | 59 | 59 | 67 | 66 |
| | ObjNet | 57.7 | 57 | 54 | 54 | 52 | 43 | 56 | 47 | 53 | 58 | 64 | 74 | 59 | 59 | 68 | 67 |
| | IN-Cartoon | 56.2 | 57 | 54 | 54 | 50 | 41 | 53 | 43 | 51 | 55 | 61 | 76 | 58 | 59 | 68 | 65 |
| | IN-Drawing | 59.7 | 61 | 59 | 59 | 51 | 45 | 56 | 46 | 55 | 63 | 66 | 77 | 59 | 61 | 69 | 67 |

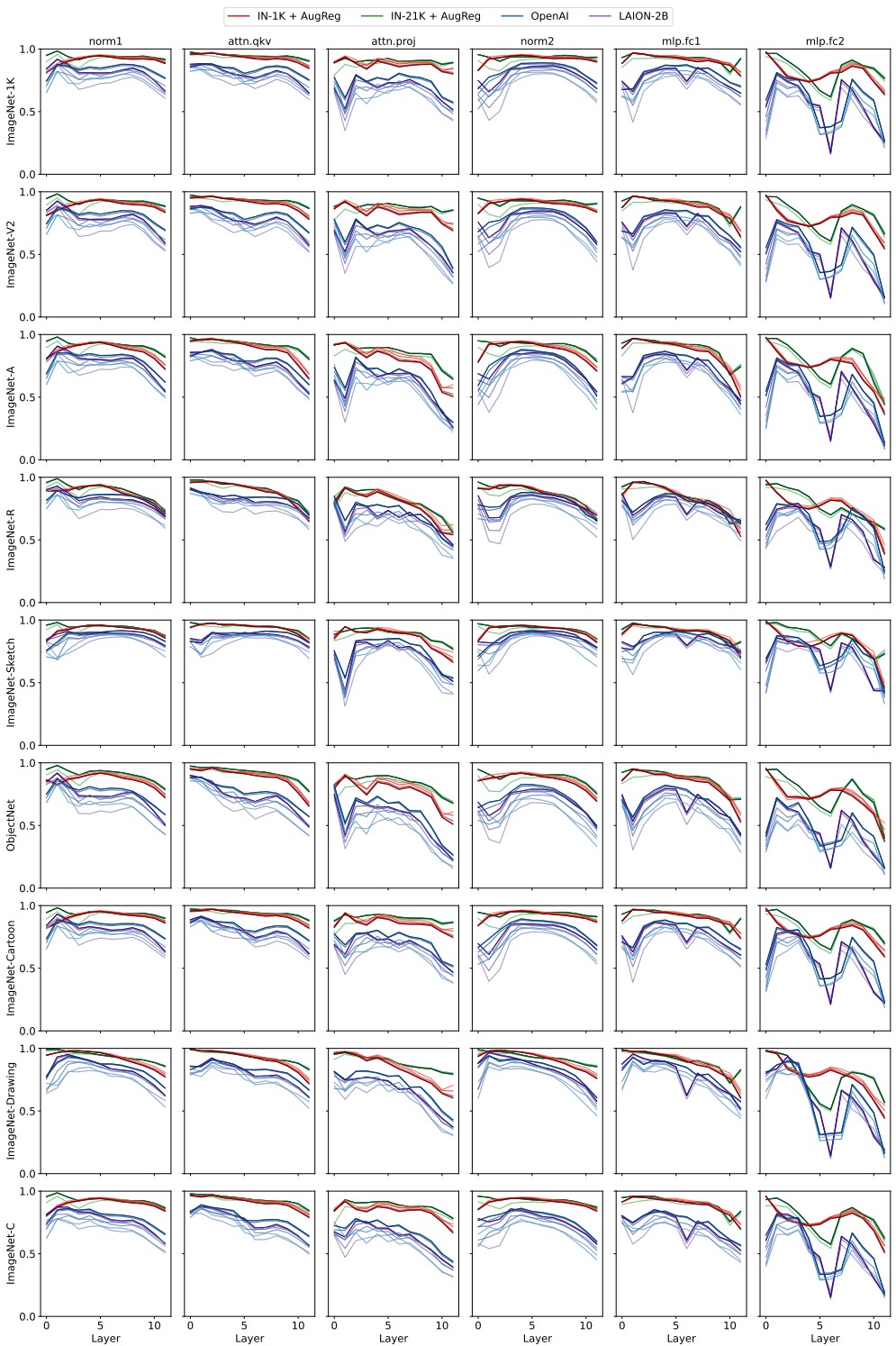

Figure 14: **Representational shifts while fine-tuning on ImageNet-R, analyzed by Centered Kernel Alignment (CKA), in ViT-B/16 models pretrained on various datasets.** Rows indicate datasets where CKA is measured and column indicates different part of transformer demonstrated in Figure 9. Darker and higher-contrast shades denote later epochs in the fine-tuning process. ImageNet-R (downstream), ImageNet-1K (pretraining), and ImageNet-A (OOD dataset) are used as evaluation datasets. Models pretrained on OpenAI and LAION-2B show distinct CKA patterns across layers compared to others.

