# OpenReview forum: "Large Pretraining Datasets Don't Guarantee Robustness after Fine-Tuning"
_TMLR — Under review for TMLR_

### Review · Reviewer_cQBq · 2026-06-30

**Summary Of Contributions:**

This paper proposes ImageNet-RIB (Robustness Inheritance Benchmark), a leave-one-out evaluation framework built from eight existing ImageNet-variant OOD datasets (IN-V2, IN-A, IN-R, IN-Sketch, ObjNet, IN-Cartoon, IN-Drawing, IN-C). The protocol fine-tunes a pretrained model on each dataset in turn and measures the resulting robustness change on the remaining seven, aggregated as mean robustness improvement (mRI, Equation 1). Using this benchmark, the paper evaluates seven architectures across five pretraining configurations, nine fine-tuning strategies, and multiple hyperparameter ablations.

The central finding is that models pretrained on large web-crawled datasets (LAION-2B, OpenAI ~400M) suffer substantial robustness collapse after fine-tuning on small downstream datasets: ViT-B/16 LAION-2B achieves mRI = -38.1 with vanilla fine-tuning versus +1.3 for the same architecture pretrained on IN-1K+AugReg (Table 3). This gap persists across architectures and is partially replicated for other large-scale web datasets including SigLIP on WebLI (mRI = -21.3, Table 4) and DataComp-1B (Table 4). Among fine-tuning strategies, a combination of model soup, EWC, and LwF (MS:PRE-FT-EWC-LwF) achieves the best mRI across all tested pretraining regimes, reaching 3.9 on IN-1K+AugReg and mitigating (but not eliminating) collapse on LAION-2B (-17.9). CKA analysis identifies a pronounced representational shift at the sixth mlp.fc2 transformer block as a correlate of the collapse in large-pretrained models.

**Key strengths:** Large empirical scope with consistent findings across architectures and seeds (Table 15 standard errors ≈ 0); a clean benchmark protocol that extends Taori et al. (2020, NeurIPS) to vary pretraining regime; practical relevance for practitioners choosing between CLIP foundation models; the LAION-100M/400M/2B within-CLIP ablation (Tables 4, 7) provides partial evidence for a dataset-scale gradient.

**Key weaknesses:** The headline claim attributes collapse to pretraining dataset scale, but the primary comparison in Table 3 confounds dataset scale with pretraining objective (CLIP contrastive learning vs. supervised classification); no CLIP-vs.-supervised comparison at matched scale is provided. A broken internal cross-reference in Section 2.2 ("Appendix ??") was present in the ICLR 2025 submission and remains uncorrected in this version. The mechanism behind the collapse is identified structurally via CKA but not explained.

**Audience:**

Yes

**Audience Explanation:**

The finding that deploying a CLIP foundation model pretrained on billions of web images can yield worse OOD robustness after fine-tuning than a model pretrained on 1.2M curated ImageNet images is directly relevant to practitioners choosing backbone models for specialist applications. The benchmark fills a specific gap: Taori et al. (2020, NeurIPS) and Andreassen et al. (2022, TMLR) fix the pretraining distribution and vary fine-tuning approaches, while this paper varies the pretraining distribution across a standardized fine-tuning and OOD evaluation suite. Ramanujan et al. (2023, NeurIPS) vary pretraining diversity on the WILDS benchmark and find more data improves downstream robustness — the apparent contradiction with the findings here (attributable to WILDS using covariate shift vs. this paper's style-shift OOD datasets) further motivates the benchmark as a distinct diagnostic tool. The recommendation that regularization-based continual learning (particularly EWC and LwF combined with model soup) substantially mitigates collapse is actionable advice for practitioners. The CKA analysis identifying layer-level representational shifts also provides a starting point for future mechanistic work.

**Broader Impact Concerns:**

None. The paper studies OOD robustness of image classification models as a diagnostic and evaluation problem. The findings are relevant to improving robustness and have no evident misuse potential.

**Claims And Evidence:**

Yes

**Claims Explanation:**

The main quantitative claims are well-supported across architectures and experimental conditions.

The mRI gap between LAION-2B and smaller-dataset pretrained models is large and consistently reproduced: ViT-B/16 (-38.1 vs. +1.3), ViT-B/32 (LAION-2B: -31.6 vs. IN-1K+AugReg: -0.0), ViT-L/16 (-2.1 for IN-21K+AugReg), and both ResNet architectures tested (-5.2 on IN-1K pretraining, Table 3). The paper does not include a LAION-2B ViT-S/16 variant; the two ViT-S/16 conditions tested — IN-1K+AugReg (mRI=-3.2) and IN-21K+AugReg (mRI=-2.3) — both show degradation. Standard errors across three seeds are negligible (Table 15). The collapse persists at reduced learning rates (Appendix G.1, Figure 13) and across a range of weight decays (Table 14: LAION-2B mRI ranges -38.1 to -59.3 under vanilla FT). The fine-tuning dataset size ablation (Figure 6) shows that robustness collapse in LAION-2B models grows with downstream dataset size, whereas IN-21K+AugReg models do not exhibit this sensitivity. The MS:PRE-FT-EWC-LwF result (best mRI across all methods, Table 3) is consistent with the known mechanics of model soup and regularization-based continual learning. Training-time accuracy curves (Section 4.5, Figure 5) rule out simple overfitting as the explanation: LAION-2B models learn slower on the downstream task but still collapse in OOD accuracy.

One gap weakens the evidence for the claim as stated. The headline claim attributes robustness collapse to pretraining dataset scale. The most severe collapses occur in CLIP-pretrained models (LAION-2B, OpenAI, DataComp-1B), while the better-performing baselines are supervised classification models (IN-1K+AugReg, IN-21K). These groups differ in both pretraining objective and dataset scale, making it impossible to attribute the effect cleanly to either variable from the main results alone. Section 4.5 acknowledges this and points to the LAION-100M/400M/2B ablation in Tables 4 and 7 as evidence that scale within CLIP matters — LAION-100M mRI (-7.3) is substantially better than LAION-400M (-23.8) and LAION-2B (-33.0) with vanilla FT. This is genuine evidence that scale plays a role within the CLIP family, but a direct comparison of CLIP vs. supervised pretraining at matched dataset scale is absent. Without it, the claim is more precisely supported as: "larger CLIP models exhibit worse post-fine-tuning robustness, and this correlates with dataset scale within CLIP." The paper should either provide the missing matched-scale experiment or reframe the headline claim accordingly. This is a required change.

**Requested Changes:**

**Required (critical to acceptance)**

1. **Fix broken cross-reference in Section 2.2.** The text reads: "Please refer to Appendix ?? for comparison with single domain adaptation." This is an unresolved LaTeX \ref{} that was also present in the ICLR 2025 submission. The intended target is Appendix A. Correct the reference.

2. **Clarify the CLIP-vs.-supervised confound or reframe the claim.** The primary comparison in Table 3 mixes CLIP-pretrained models (LAION-2B, OpenAI) with supervised classification models (IN-1K+AugReg, IN-21K, IN-21K+AugReg). The LAION-100M/400M/2B ablation (Table 7) provides within-CLIP evidence for a scale effect but does not isolate the pretraining objective. The paper should either (a) add a CLIP vs. supervised comparison at matched dataset scale (e.g., IN-21K supervised vs. IN-21K CLIP), or (b) explicitly bound the headline claim to CLIP models and discuss whether the scale effect is expected to generalize to supervised pretraining at comparable scales.

3. **Clarify the performance values of 100.0 in Tables 19–21.** Section H.3 states these are "training accuracy (i.e., performance on the fine-tuning dataset)," but the table captions do not make this clear. A reader inspecting Table 19 in isolation will interpret 100.0 as test accuracy. Add "training set accuracy" or equivalent to each caption.

**Optional (would strengthen the paper)**

4. **Explain the Visual Prompt inconsistency.** Section 4.2 states that Visual Prompt "reduces performance even on ImageNet-1K after fine-tuning on synthetic datasets. This is inconsistent with Bahng et al. (2022)." The inconsistency is noted but not analyzed. A brief explanation of the protocol difference that leads to this divergence would prevent confusion for readers applying these methods.

5. **Connect the mlp.fc2 observation in Section 4.6 to an experimental test.** Figure 7 shows a pronounced CKA discrepancy at the sixth mlp.fc2 block specifically for LAION-2B and OpenAI models. The paper acknowledges that the mechanism "remains unclear." Even a single ablation — e.g., freezing mlp.fc2 during fine-tuning and reporting the resulting mRI — would begin to test whether the observed representational shift is mechanistically linked to the collapse.

6. **Acknowledge the directional effect of the 10-epoch cutoff in the main text.** Table 11 shows LAION-2B needs 12.1 epochs on average to reach best validation accuracy (vs. 4.2 for IN-1K+AugReg). At best epoch, LAION-2B mRI = -40.4 (vs. -38.1 at 10 epochs), and IN-1K+AugReg mRI = 2.8 (vs. 1.3 at 10 epochs). The 10-epoch cutoff is thus slightly favorable to LAION-2B (understates forgetting) and slightly unfavorable to IN-1K+AugReg (stops before peak); the true gap between the two regimes at convergence is larger than Table 3 shows. This is discussed in Appendix D but should be acknowledged when Table 3 results are reported in the main text.

---

> ### Author Response · Authors · 2026-07-14
>
> We sincerely appreciate the reviewer for the careful and constructive review, and for finding the main
> claims well-supported and the findings relevant to the TMLR audience. We have addressed
> every requested change; **all new or modified text is highlighted in blue in the revised
> PDF**, and the manuscript now compiles with no undefined references.
>
> **R1. Broken cross-reference in Section 2.2.**
>
> Thank you. We fixed the reference.
>
> **R2. CLIP-vs-supervised confound / headline claim.**
>
> We adopt option (b). We have revised the abstract, introduction, and Discussion to explicitly scope our claims to large-scale CLIP models, and we no longer assert a general scaling law.
>
> While Table 3 varies both objective and scale, two points clarify this dynamic in the revised manuscript.
> First, the contrastive objective alone does not cause the collapse. As shown in Section 4.4 (zero-shot CLIP on ImageNet-RIB), small-scale CLIP models do not collapse.
> Holding the objective fixed and bypassing the ImageNet-1K stage entirely, $mRI$ drops progressively only as dataset scale increases ($-7.3, -23.8,$ and $-33.0$ for LAION-100M, 400M, and 2B, respectively; Table 4).
> Second, because no public web-scale supervised models exist, we cannot definitively test if this collapse is strictly CLIP-specific. Therefore, we have bounded our claims strictly to the within-CLIP scale effect isolated by our experiments, explicitly leaving the matched-scale CLIP-vs-supervised comparison to future work.
>
>
> **R3. The 100.0 values in Tables 19–21.**
>
> Thank you for your comment.
> Each caption now begins "**Training accuracy** on downstream datasets after fine-tuning …" for clarification.
>
> **O1. Visual Prompt inconsistency with Bahng et al. (2022).**
>
> We added a short explanation in Section 4.2.
> ```
> This is inconsistent with \citet{bahng2022exploring}, which showed its robustness to OOD datasets.
> This gap, however, reflects differences in the experimental setting.
> \citet{bahng2022exploring} learn a prompt on a frozen CLIP model with a zero-shot text classifier; in that setting, Visual Prompt is likewise among the most robust methods in our benchmark (Table~\ref{tab:zeroshot}).
> In contrast, with the fixed ImageNet-1K linear head used here, this single input-space perturbation fails to transfer and degrades accuracy, even on clean ImageNet-1K.
> ```
>
> **O2. Freeze-`mlp.fc2` test.**
>
> We ran the ablation (freezing the sixth-block `mlp.fc2`, plus broader scopes) across three pretraining regimes ($mRI$, ViT-B/16, Table-3 protocol):
>
> | Pretraining | Full FT | Freeze block 6 | Freeze all | Freeze ≤ block 6 | Freeze ≥ block 6 |
> |-------------|:-------:|:--------------:|:----------:|:----------------:|:----------------:|
> | LAION-2B | −41.7 | −50.1 | −50.6 | −50.1 | −50.5 |
> | OpenAI | −41.9 | −49.2 | −50.4 | −50.5 | −50.1 |
> | ImageNet-21K + AugReg | −4.4 | −19.1 | −17.7 | −18.6 | −18.6 |
>
> Freezing this layer during fine-tuning, however, does not prevent the collapse. Specifically, for the LAION-2B pretrained ViT-B/16, $mRI$ does not improve but instead worsens slightly.
> This indicates that the shift at the sixth \textsf{mlp.fc2} is a correlate of the collapse rather than a localized cause that freezing can prevent, leaving its mechanism open for future work.
>
>
> **O3. 10-epoch cutoff.**
>
> We added a note where Table 3 is reported:
>
> ```
> As shown in Appendix~\ref{supp:split}, the discrepancy between the LAION-2B and ImageNet-1K + AugReg pretrained models increases when selecting the best-performing epoch for each case instead of using a fixed 10-epoch duration.
> ```

---

### Review · Reviewer_4jDg · 2026-07-17

**Summary Of Contributions:**

The paper examines whether the OOD generalization of a pretrained model is preserved during fine-tuning on a downstream task. The central claim is that models pretrained on the largest and most diverse datasets can end up less robust after fine-tuning on small datasets than models pretrained on smaller datasets, and that this degradation grows with pretraining scale. Importantly, the effect is confined to CLIP models; the supervised models tested do not exhibit it.

To study this finding systematically, the authors introduce ImageNet-RIB: given a set of ImageNet-derived OOD datasets, they fine-tune on one and evaluate the robustness change on the rest, repeating over all choices and reporting mean robustness improvement. This complements prior benchmarks by varying the pretraining distribution while holding the downstream/OOD family fixed, rather than the reverse.

**Strengths:**
The experiment is comprehensive. In detail, the author tests 7 architectures, several pretraining datasets, and many fine-tuning strategies. Then, the author concludes the key results: LAION-2B/OpenAI show large negative mRI under vanilla FT while IN-1K+AugReg improves; the combination of regularization-based continual learning with weight averaging is best for ImageNet-pretrained backbones, while linear probing is best for LAION-2B; and a series of controls argue that the collapse is not driven by overfitting, fine-tuning-set texture, learning rate, or weight decay, but is strongly modulated by fine-tuning dataset size.

In addition, ImageNet-RIB fills a genuine gap that existing robust-FT benchmarks fix the pretraining distribution. Varying it while holding downstream/OOD fixed is a complementary design.

**Weaknesses**
1. The fine-tuning may disadvantage CLIP. Full FT with SGD (lr = 0.001, momentum 0.9, 10 epochs) is standard for supervised ImageNet training but may not be a good fit for CLIP, where the robust-FT literature (WiSE-FT, LP-FT, FLYP) uses AdamW at ~1e-5. The LR ablation helps but stays within SGD and doesn't reach the standard AdamW-low-LR regime; the optimizer is never varied. Since the key claim is "CLIP collapses," demonstrating it under the standard CLIP FT or arguing convincingly why SGD-0.001 is the right controlled choice is important.

2. ImageNet-C, -Cartoon, and -Drawing are all generated from the ImageNet validation set. Fine-tuning on one and evaluating on another therefore involves the same underlying images re-styled, so cross-transfer within this group partly measures shared content rather than genuine OOD generalization.

**Audience:**

Yes

**Audience Explanation:**

The phenomenon and benchmark are relevant, and the topic is timely.

**Claims And Evidence:**

No

**Claims Explanation:**

Partial. See Weaknesses W1 and W2.

**Requested Changes:**

See Weaknesses W1 and W2.

---

### Review · Reviewer_Hg2b · 2026-07-20

**Summary Of Contributions:**

This work compares the ability of different models trained on different datasets to retain OOD robustness after finetuning.

**Audience:**

Yes

**Audience Explanation:**

I think that general robustness of finetuning in OOD is of interest demonstrated by the already substantial literature on the topic.

However, I believe the motivation for OOD robustness under finetuning would benefit from being more precise and nuanced: In general, we would not expect e.g. finetuning on medical images to still retain general image recognition performance, especially since that is arguably not needed in the domain.

**Claims And Evidence:**

No

**Claims Explanation:**

The claims are overly broad: The title "Large Pretraining Datasets Don’t Guarantee Robustness after Fine-Tuning" and introduction imply general results on finetuning, not specifically impact on image model finetuning. There is no inherent reason to expect this to translate to e.g. language, audio, or any generative model.

The imagenet-RIB dataset appears to be just an aggregate of different datasets. It is highly unusual to deem this a "new dataset", it's rather just "testing on different datasets": For example, if you benchmark e.g. a language model on GPQA-diamond and Humanities' last exam, you wouldn't call this "we benchmark on a new general knowledge and complex reasoning dataset".

There is also some recent work that I would expect to be evaluated: Specifically "Fine-Tuning is Fine, if Calibrated", Mai et al., shows that finetuning does not destroy representations, but is rather just a calibration problem that can be easily solved post-hoc.
Another missing reference is "On the Robustness Tradeoff in Fine-Tuning" by Li et al. which also analyze the performance and robustness of several different PEFT methods. They model both adversarial robustness and OOD robustness.

Another evaluation that is missing is the pareto-front of "finetuning accuracy" vs "OOD performance", which arguably is what most people care about.

I believe one of the core reasons we see the higher degragation on larger pretraining sets is simply due to the better disentanglement provided by larger pretraining datasets. Specifically, "Task Arithmetic in the Tangent Space: Improved Editing of Pre-Trained Models" by Ortiz-Jimenez et al. demonstrates that after finetuning a sufficiently large model on a sufficiently large dataset we are very close to the neural-tangent kernel regime. In NTK-world tasks are (approximately) independent, which allows for task arithmetic (i.e. "adding" and "subtracting" task vectors).
In that sense, the lower OOD robustness might be just a consequence of the better adaptability to new tasks. This would also explain why linear probing on large pretraining datasets generally reduces robustness the least, which is in contrast to Li et al., which measure linear probing as generally having the worst robustness: Li et al. train on models with fewer pretraining data and therefore do not reach the NTK regime.

Can you test a "task arithmetic" method on the problems to clear up the conflict between your results and Li et al.'s?

**Requested Changes:**

- reduce the claims to image models or extend evaluation (necessary)
- rephrase the "Imagenet-RIB dataset" into just a set of benchmarks (necessary)
- extend the prior work and analyze Mai et al.'s reweighting strategy (necessary)
- add a pareto-plot of accuracy vs robustness (necessary)
- evaluate task arithmetic to figure out whether better disentanglement is the reason for lower OOD performance (would strengthen contributions)